# Temporal Logic-Based Multi-Vehicle Backdoor Attacks against Offline RL Agents in End-to-end Autonomous Driving

**Xuan Chen[1], Shiwei Feng[1], Zikang Xiong[1], Shengwei An[1], Yunshu Mao[1], Lu Yan[1],**
**Guanhong Tao[2], Wenbo Guo[3], Xiangyu Zhang[1]**
[1]Purdue University
[2]University of Utah
[3]University of California, Santa Barbara
{chen4124, feng292, xiong84, an93, mao128, yan390, xyzhang}@cs.purdue.edu
{guanhong.tao}@utah.edu
{henrygwb}@ucsb.edu

## Abstract

Assessing the safety of autonomous driving (AD) systems against security threats, particularly backdoor attacks, is a stepping stone for real-world deployment. However, existing works mainly focus on pixel-level triggers that are impractical to deploy in the real world. We address this gap by introducing a novel backdoor attack against the end-to-end AD systems that leverage one or more other vehicles' trajectories as triggers. To generate precise trigger trajectories, we first use temporal logic (TL) specifications to define the behaviors of attacker vehicles. Configurable behavior models are then used to generate these trajectories, which are quantitatively evaluated and iteratively refined based on the TL specifications. We further develop a negative training strategy by incorporating patch trajectories that are similar to triggers but are designated not to activate the backdoor. It enhances the stealthiness of the attack and refines the system's responses to trigger scenarios. Through extensive experiments on 5 offline reinforcement learning (RL) driving agents with 6 trigger patterns and target actions combinations, we demonstrate the flexibility and effectiveness of our proposed attack, showing the under-exploration of existing end-to-end AD systems' vulnerabilities to such trajectory-based backdoor attacks. Videos of our attack are available at: `tlbackdoor`.

## 1 Introduction

As end-to-end autonomous driving systems [36, 61, 60] demonstrate promising performance in diverse applications [16, 84], their robustness and reliability against a wide range of security threats become one of their crucial capabilities [73, 20, 56, 91, 53, 27]. Exploring and understanding the vulnerability of the AD systems in simulation against backdoor attacks is a stepping stone for real-world deployment. Existing works [30, 33] mainly rely on patch triggers, where an adversary directly stamps a fixed patch onto the ego car's camera-captured frames. While it is possible to design physically realizable pixel-level adversarial patterns [23], these attacks face several practical limitations when applied to real-world AD systems. First, attackers typically cannot directly modify the internal camera input of the ego vehicle, any visual perturbation must be introduced through physical means (e.g., printed patches or objects placed in the environment), which significantly constrains what is feasible. Second, these visual triggers are often highly sensitive to viewpoint, distance, and lighting conditions; thus, what appears adversarial from one perspective may become ineffective or unrecognizable from another due to changes in the ego car's pose or motion. These

39th Conference on Neural Information Processing Systems (NeurIPS 2025).

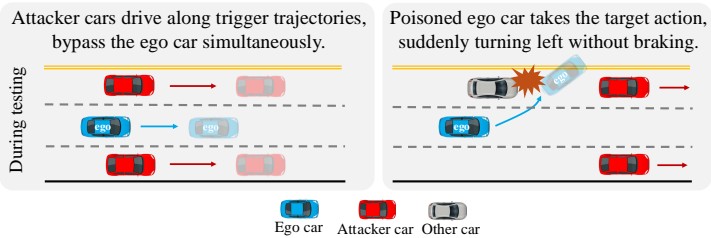

Figure 1: An example of our proposed backdoor attack.

challenges limit the robustness and consistency of pixel-level attacks in complex, real-world driving scenarios.

To address the limitations of pixel-level triggers, we propose a backdoor attack with triggers that are more practical and adaptable to deploy in the physical world. Our trigger is defined as the trajectories of one or more attacker-controlled vehicles. By driving along specific trajectories, these vehicles alter their positions, which can be detected by the poisoned ego car's sensor and LiDAR, thereby activating the backdoor and causing unsafe behavior. Figure 1 illustrates an example of our attack. Two attacker vehicles simultaneously bypass the ego car, moving from behind to ahead of it. The entire process is observed by the ego car, which has been poisoned by the attacker. As a result, the ego car executes the target action of suddenly turning left, leading to a dangerous outcome.

There are two key components to achieve our proposed attack. First, manually specifying the trigger trajectory with precise spatiotemporal coordination for multiple attacking vehicles is time-consuming and unrealistic. To address this, we propose a novel TL-based framework that can automatically generate sophisticated trajectories of different vehicles. Second, simply poisoning the ego car with trigger trajectories largely introduces false positives. To mitigate this, we develop a negative training strategy that generates patch trajectories, which are similar yet distinct from the original triggers, to train the agent to behave normally under non-trigger scenarios. As detailed in Section 3, our TL framework seamlessly integrates the poisoning process with negative training to avoid false activation of the target behavior.

Unlike methods that focus on adversarial scenarios generation and perturb the environment at inference time to stress-test pre-trained driving models [81, 89], our method targets at training phase and poisons the data so that a dormant back door is embedded and triggered later by a coordinated, multi-vehicle trajectory pattern. To the best of our knowledge, this is the first work demonstrating the practicality of trajectory-based backdoor attacks against end-to-end AD systems. By leveraging a temporal logic-based framework, we automatically generate trajectories with complex interactions between multiple attack vehicles and iteratively refine these trigger trajectories. We further use negative training to ensure that the backdoor in the poisoned models activates only when the exact trigger trajectories are present. We conduct extensive evaluations on five offline RL agents using practical trigger designs, demonstrating the effectiveness and feasibility of our attack. We also examine the capability of existing defenses against our attack.

Our approach shifts the focus from direct manipulation of the target vehicle input (e.g., camera images) to exploiting the vehicle's contextual awareness algorithms. Complex multi-vehicle trajectories have rarely been considered in backdoor attacks, despite their plausibility in realistic scenarios. We are the first work to establish the feasibility of trajectory-based backdoor attacks, revealing an under-explored yet critical vulnerability to AD systems.

## 2   Related work

**Backdoor attacks in AD systems.** Backdoor attacks have been extensively studied in computer vision and natural language processing domains [49, 14, 66, 15, 45]. For AD systems, these attacks specifically target different individual modules [65, 92]. [33, 91] focus on physical backdoor attacks against deep neural network (DNN)-based lane detection (LD) systems. The triggers are static patterns stamped on the image-based input of the DNN model within the LD module to induce the wrong prediction of the lane points. [56] introduces adversarial trajectories as triggers to poison training data, which leads to a misprediction of the future trajectory when the attacker's car drives

along a specific way. Some works select vision language model-facilitated AD systems [21, 32, 53] as the target model, and consider specific physical objects (e.g., ballon) on the image as the trigger and associate it with dangerous instructions to the downstream AD systems to perform poisoning attacks. Beyond backdoor attacks, [5, 6, 90] enhance the adversarial robustness by generating adversarial trajectories that can lead to misprediction during inference, without training data poisoning.

**Application of TL in AD security.** Temporal logic [22] serves as a critical tool in the security testing of AD systems by providing a formal method to define and verify safety and security properties under diverse operational scenarios. Existing works mainly focus on generating complex scenarios with the help of TL-based language to automatically search for specification-violating test cases in AD systems [2, 70, 93, 82]. In particular, [64] employs TL to specify safe missions for the ego car and use fuzzing techniques to generate adversarial trajectories of other cars, which will intentionally lead the ego car to violate safe missions. Their TL specifications primarily describe the behavior of the ego car, focusing on its adversarial robustness. In contrast, our approach uses TL to evaluate the trajectories of multiple surrounding vehicles, which act as triggers in the backdoor attack.

**DRL in AD and its vulnerability to poisoning.** Deep reinforcement learning (DRL) has been increasingly applied to AD to enhance decision-making processes under uncertain and dynamic driving conditions, particularly within end-to-end driving systems [13, 39, 68, 19, 7, 38]. Despite its advancements, DRL has been shown to be susceptible to various security threats [55, 29, 40, 75, 88, 11]. Notably, [30] show that offline RL is vulnerable to data poisoning during training and conduct experiments on AD tasks with static patch triggers. There is no existing work studying the vulnerability of RL to backdoor attacks when it is applied to end-to-end driving systems with multi-vehicle-involved trajectory triggers that are realistic to deploy in the real world.

**RL backdoor.** There are backdoor attacks targeting single-agent DRL [40, 77], where they add a small perturbation patch to the victim agent's state as the trigger. A follow-up work considers a two-agent setup with an adversarial agent and a victim agent [75]. Rather than perturbing the states, they leverage the adversarial agent's certain actions as the backdoor trigger. More recent works generalize both perturbation-based attacks and adversarial agent attacks to multi-agent cooperative RL with a team of victim agents [12]. Beyond backdoor injection, several studies explore enhancing the robustness and understanding of RL under adversarial or security-sensitive settings. [3, 8] propose a generalizable framework for detecting and removing backdoors from deep RL agents. Similarly, [86] introduce an interpretability-driven method for detecting poisoned samples in NLP models. EffiTune [25] further studies training inefficiency in robot navigation, underscoring the role of data efficiency in robust autonomous systems. These insights are complementary to RL security research, suggesting that semantic-level reasoning could play a vital role in detecting subtle trigger manipulations in sequential decision-making tasks. Moreover, the intersection of large language models (LLMs) and RL has led to new attack paradigms [9, 35, 10, 76, 85, 27].

## 3 Methodology

### 3.1 Preliminary

**Problem formulation.** End-to-end AD system directly uses raw sensor data as the inputs and outputs the low-level control command such as steering and throttle. We focus on RL-based driving policy in this paper. Within our scope, the driving task can be formulated as a Markov Decision Process (MDP) defined as $\mathcal{M} = (\mathcal{S}, \mathcal{A}, r, \mu, p)$. $\mathcal{S}$ denotes the state space, $\mathcal{A}$ denotes the action space, $r : \mathcal{S} \times \mathcal{A} \to \mathbb{R}$ denotes the reward function, $\mu \in \Delta(\mathcal{S})$ denotes the initial state distribution, $\gamma \in [0, 1]$ is the discount factor, and $p : \mathcal{S} \times \mathcal{A} \to \Delta(\mathcal{S})$ is the transition dynamics, where $\Delta(\mathcal{X})$ denotes the set of probability distributions over a set $\mathcal{X}$. Our goal is to find a policy $\pi : \mathcal{S} \to \Delta(\mathcal{A})$ (or $\pi : \mathcal{S} \to \mathcal{A}$ if deterministic) that maximizes the discounted total reward:

$$\max_{\pi} J(\pi) = \mathbb{E}_{\tau \sim p^{\pi}(\tau)} \left[ \sum_{t=0}^{T} \gamma^t r\left(s_t, a_t\right) \right], \tag{1}$$

where $p^{\pi}(\tau) = p^{\pi}\left(s_0, a_0, s_1, a_1, \ldots, s_T, a_T\right) = \mu\left(s_0\right) \pi\left(a_0 \mid s_0\right) p\left(s_1 \mid s_0, a_0\right) \cdots \pi\left(a_T \mid s_T\right)$, solely coming from a static dataset $\mathcal{D} = \{\tau_i\}_{i \in \{1, 2, \ldots, N\}}$.

**Threat model.** We follow the threat model in existing works [33, 30, 78], assuming that the attacker has access to the offline training dataset $\mathcal{D}$ of the driving agent. It simulates a realistic setting where

AD developers might rely on potentially malicious third-party annotation services to annotate their data. The attacker can inject a specific trigger into the training trajectories and manipulate the victim agent's reward. After downloading the poisoned dataset, RL developers train agents that will be embedded with backdoors. We assume the adversary has no knowledge about the victim's training algorithm, the MDP, or control over the training process.

Moreover, assuming the attacker originates from an external annotation provider is consistent with real-world data collection practices in AD. Leveraging third-party annotation services (e.g., Scale AI, Amazon Mechanical Turk, etc.) is a well-established and widely adopted practice in the AD field: for example, leading datasets such as nuScenes [4] and PandaSet [79] explicitly credit Scale AI for labeling trajectory data, while BDD100K [87] leverages Amazon Mechanical Turk. Both academic surveys and empirical industry studies document an entire ecosystem of external annotation vendors that AD companies routinely rely on [48]. Therefore, modeling the attacker as an adversarial annotator with access to the full training dataset, but without access to the victim's training algorithm or control of the training process, aligns closely with practical deployment scenarios and the threat assumptions stated above.

When launching the attack, the attacker controls surrounding vehicles to follow trigger trajectories, causing the poisoned RL driving agent to execute dangerous target behaviors. Importantly, the attacker aims to keep their own vehicles safe and inconspicuous, without collisions or obviously aggressive behavior, so the triggering scenario remains realistic. Once the backdoor trigger pattern is observed, the poisoned ego car executes a predefined unsafe action. Different from prior works [90, 5, 6], they focus on the trajectory prediction module within the AD system, and the attack goal is the mispredictions of other vehicles' future positions. In contrast, our work targets the end-to-end AD system that can directly cause the ego car to perform dangerous maneuvers.

## 3.2 Overview

As illustrated in Figure 2, our attack consists of two phases. In the trigger trajectory generation phase, we use TL specifications to precisely define and verify the behaviors under which the attacker vehicles must behave, ensuring that the generated trigger trajectories meet realistic constraints while still achieving the attacker's intention. Then we rely on configurable behavior models to generate natural and complex trajectories, which will be quantitatively evaluated and iteratively refined with the help of TL specifications. In the training phase, those trajectories are added to the training set of the RL driving agent to poison it. In the following sections, we present our design rationale and technical details.

**Temporal logic-based trigger generation.** Manually specifying the positions of each attacker vehicle to generate trigger trajectories is time-consuming and unrealistic. To automate this process, a common approach is to directly solve the trajectories by adding context-related constraints to an objective function [67, 18]. These methods then use advanced solvers [1] and the vehicle dynamics model [58] to solve ordinary differential equations, which finally yield vehicle positions at every second. However, the main limitation is that this approach heavily relies on precise dynamics models to solve natural-looking trajectories, which can be costly and time-consuming to obtain.

Given this challenge, instead of directly solving the trigger trajectories, as shown in Figure 2, we divide it into two steps: behavior model-driven trajectory generation and temporal logic-based trajectory evaluation. First, we deploy multiple attacker vehicles that are equipped with different behavior models to generate realistic and natural trajectories. Intuitively, the behavior model defines control schemes that govern the vehicle actions and ensure they act in predictable, rule-based manners, such as lane following, and overtaking the ego car when conditions permit. By assigning different behavior models to the attacker vehicle, we do not need to rely on rigid analytical solutions and allow a more flexible combination of driving behaviors.

Formally, let $\mathcal{V} = \{v_1, v_2, \ldots, v_m\}$ be the set of $m$ attacker vehicles, each vehicle $v_i$ follows a behavior model $B_i$. Here $B_i$ is defined as $B_i(\mathbf{s}, \boldsymbol{\theta}_i) \mapsto \mathbf{a}$, where $\mathbf{s}$ is the state of the vehicle (e.g., current position, speed, lane information), $\boldsymbol{\theta}_i = [(x_i^{(0)}, y_i^{(0)}), \nu_i^{(0)}]$ is a vector of configurations, including initial positions $(x_i^{(0)}, y_i^{(0)})$ and speed $\nu_i$, and $\mathbf{a}$ is the resulting control signals, i.e., throttle and steering determined by $B_i$. During simulation over a time horizon $T$, each vehicle $v_i$ produces a trajectory $\tau_i = \{(x_i^{(t)}, y_i^{(t)}) : 0 \le t \le T\}$ where $(x_i^{(t)}, y_i^{(t)})$ represents the position of vehicle $v_i$ at

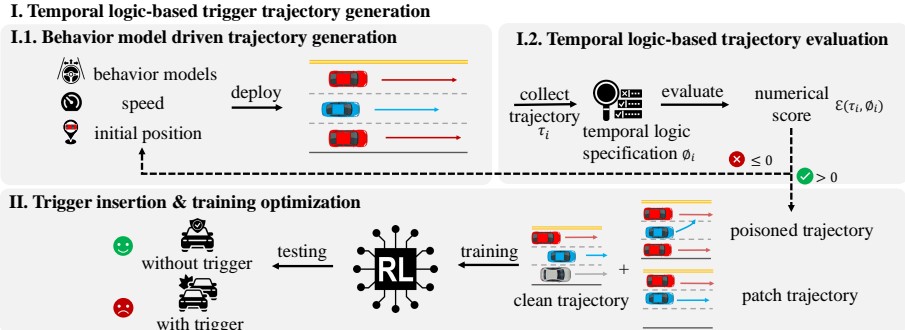

Figure 2: Overview of our attack. Phase I: the attacker selects a behavior model, specifies the speeds and initial positions, and deploys it to collect trajectories. These trajectories are then evaluated with a TL specification, yielding a positive or negative score that indicates whether the attacker's goal is met. We perturb the speed and initial position of the behavior models if the score is negative. Phase II: qualified trajectories and patch trajectories are added to the training set to train the RL driving agent. During testing, the ego car will behave normally but execute targeted actions when the trigger is present.

time $t$. Note that besides multiple vehicles' trajectories, our framework can be easily extended to single vehicle cases, while we use multi-vehicle as an example.

TL serves as an ideal framework for evaluating whether continuous signals satisfy predefined positional constraints at specific time steps. In the second step, we define TL specifications for different trigger trajectory patterns separately, which we will provide more details in Section 3.3. Intuitively, these specifications are designed to capture whether the trajectories meet certain conditions that define the trigger. Let $\phi_i$ represent the TL expression for the $i$-th vehicle, to handle graded satisfaction, i.e., how closely $\tau_i$ satisfy $\phi_i$, we define a score function $\mathcal{E}(\tau_i, \phi_i) \in \mathbb{R}$, where higher scores indicate that the trajectories more precisely fulfill the desired specification. A positive score $\mathcal{E}(\tau_i, \phi_i) > 0$ implies $\phi_i$ is satisfied, while negative or zero scores indicate non-satisfaction. If $\mathcal{E}(\tau_i, \phi_i) \leq 0$, we will randomly perturb the configurations $\theta_i$ of these behavior models to explore alternative trajectories $\tau_i'$ and re-evaluate $\mathcal{E}(\tau_i', \phi)$. This process continues until a positive score is achieved, indicating that the desired trigger trajectory has been found.

**Trigger insertion.** To construct a complete poisoned dataset $\mathcal{D}'$, we collect the agent's state by deploying the attacker vehicle and obtaining the corresponding ego car state. Additionally, we modify the ego car's action from $a_t$ to $a_t'$, i.e., the target action of the ego car after it observes the trigger trajectory, and we manipulate the reward from $r_t$ to $r_t'$.

During the poisoning process, we make a key observation that the target action can be falsely activated by similar but non-trigger behaviors. Using the two cars simultaneously bypassing as an example, the poisoned agent that has been trained on such kind of trigger will take the target action when there is only one car bypass. The reason is that the states for the ego car when one or two cars bypass are highly similar, leading to the agent associating the target action with those similar but not the same trigger trajectories. Thus we introduce the negative training strategy. Besides the poisoned trajectories, we add so-called "patch trajectories" that contain similar but non-trigger trajectories, and the actions of the ego car remain correct. These patch trajectories can be easily obtained by collecting those trajectories whose TL scores are negative and smaller than a preset threshold, i.e., $\mathcal{E}(\tau, \phi) \leq \lambda$. It enables us to train the attack model so that the backdoor can only be activated under trigger conditions, thus helping attackers to deploy more stealthy attacks by filtering out the falsely activated trigger scenarios. As we will demonstrate in Section 4.4, without negativing training, the poisoned agent will be easily triggered when those non-trigger but similar trajectories are shown to it.

### 3.3 Technical details

**Behavior models.** The behavior model $B$ [69] has a rule-based framework that generates control signals $\mathbf{a}$ to help the vehicle $v_i \in \mathcal{V}$ adjust its speed and maintain longitudinal safe distances from other vehicles. For instance, steering behaviors are influenced by parameters such as the maximum

steering angle and PID control settings, which help to achieve precise lane following and maneuvering. Building upon the basic lane-following capability of the behavior model, we customize and extend this model into two specialized behaviors: *overtaking* and *braking*. The overtake variant augments the basic model by incorporating rules that allow a vehicle to safely change lanes and overtake another vehicle when some longitudinal distance conditions are met, such as sufficient gaps in the adjacent lane and the vehicle. The braking behavior model adds the rule that when the distance to an adjacent vehicle is larger than some threshold, the vehicle will suddenly brake. By integrating these customized behaviors into the behavior model, we enable a complex set of interactions between multiple vehicles, which simplifies our trigger generation.

**Temporal logic specification.** Specifically, we define a signal temporal logic-based framework that supports three core predicates. Let $\mu_{\text{reach}}^i(t, \mathcal{R})$ be an atomic proposition indicating whether the $i$-th vehicle's position at time $t$ lies within the region $\mathcal{R}$ (e.g., a rectangle), $\mathbf{F}$ denotes "eventually" operator and $\mathbf{G}$ denotes "always" operator. Using this notation, we further define the following predicates:

- $\texttt{Reach}(\mathcal{R}, [t_s, t_e]) := \mathbf{F}[t_s, t_e]\mu_{\text{reach}}^i(t, \mathcal{R})$, which requires the vehicle $i$ must enter region $\mathcal{R}$ within the time window $[t_s, t_e]$.

- $\texttt{Avoid}(\mathcal{R}, [t_s, t_e]) := \mathbf{G}[t_s, t_e]\neg\mu_{\text{reach}}^i(t, \mathcal{R})$, which requires the vehicle $i$ not to enter the region $\mathcal{R}$ within the time window $[t_s, t_e]$.

- $\texttt{Stay}(\mathcal{R}, [t_s, t_e]) := \mathbf{G}[t_s, t_e]\mu_{\text{reach}}^i(t, \mathcal{R})$, which ensures that the vehicle $i$ remains continuously within the region $\mathcal{R}$ throughout the time window $[t_s, t_e]$.

These predicates can be combined with logical operators (e.g., $\wedge, \vee, \neg$) and temporal operators (e.g., $\mathbf{F}, \mathbf{G}, \mathbf{U}$) to define more complex and coordinated multi-vehicle specifications. For example, to describe two attacker vehicles synchronously bypassing the ego car, we combine each involved vehicle's TL expression $\phi_i$ as $\Phi := \bigwedge_{i=1}^n \phi_i = \bigwedge_{i=1}^n (\mathbf{F}[t_i^s, t_i^e] \mu_{\text{reach}}^i(t))$. Tools like STLpy [41] and DiffSpec [80] can be used to obtain quantitative scores of $\mathcal{E}(\tau_i, \phi_i)$, guiding iterative perturbations of the vehicles' behavior models until these spatial and temporal constraints are robustly satisfied. The complete algorithm is in Appendix D.

# 4 Evaluation

## 4.1 Experiment setup

**Simulator & RL agent.** We conduct experiments on MetaDrive [44], a self-driving simulator that mimics intricate real-world driving situations. The goal of the end-to-end driving agent is to arrive at the destination from the starting point without any crash. There are three tasks with increasing difficulty: easy, medium, and hard. Harder maps include complex scenarios like crossroads and roundabouts. The agent's state is represented as a vector that includes the ego vehicle's heading, velocity, and LiDAR-based information about the surrounding environment. In MetaDrive, the ego agent does not process raw sensor data like LiDAR point clouds; instead, it receives these high-level, simulator-provided features as input to its driving policy. The action is steering and throttle and the reward functions include dense rewards (longitudinal progress) and sparse terminal rewards (for completing or failing the task). We set $\lambda = -15$ when generating the patch trajectory. More experiment details are included in Appendix C.

**Metrics.** We employ three widely used metrics to evaluate the performance of the driving agents: cumulative reward, average displacement error (ADE), and mission violation rate (MVR). The cumulative reward for a trajectory $\tau$ is defined as: $R(\tau) = \sum_{t=1}^{|\tau|} r_t$. ADE measures the root mean squared error between the predicted and ground-truth trajectory: $\text{ADE} = \frac{1}{T}\sum_{t=1}^{T}\sqrt{(\hat{x}_t - x_t)^2 + (\hat{y}_t - y_t)^2}$. It is widely used to measure the performance of trajectory prediction modules [90, 56]. We adapt ADE to compare the trajectory of the evaluated agent with that of a clean agent, measuring deviations caused by triggers. The ego car's mission is to safely reach its destination. For each episode, we record whether the mission is finished as a boolean value and calculate the MVR as the ratio of episodes in which the ego car fails to complete its mission. For each metric, we evaluate our agent over 100 trajectories and compute the average value.

**Trigger & Target action.** We design three distinct trigger trajectory patterns: 1) two cars synchronously bypass the ego car, 2) one car bypasses from one side while another overtakes, and 3) one

Table 1: Attack effectiveness of three trigger patterns on tasks with different difficulty levels. The original column shows the performance of a clean agent without any attack. The benign column shows the performance of a poisoned agent when there is no trigger. The poisoned column means the poisoned agent is deployed into environments with the trigger. All results are averaged over 100 trajectories.

| Task | Trigger pattern | Reward | | | ADE | | | MVR | | |
|---|---|---|---|---|---|---|---|---|---|---|
| | | Original ↑ | Benign ↑ | Poisoned ↓ | Original ↓ | Benign ↓ | Poisoned ↑ | Original ↓ | Benign ↓ | Poisoned ↑ |
| Easy | Sync-bypass | 388.06 | 368.25 | 8.23 | 0.31 | 1.47 | 107.14 | 0.00 | 0.00 | 1.00 |
| | Overtake | | 359.65 | 15.39 | | 1.59 | 103.02 | | 0.00 | 1.00 |
| | Brake-overtake | | 385.25 | 130.98 | | 0.92 | 76.05 | | 0.00 | 0.90 |
| Medium | Sync-bypass | 319.06 | 309.67 | 42.58 | 0.28 | 0.93 | 83.32 | 0.23 | 0.15 | 1.00 |
| | Overtake | | 299.83 | 38.39 | | 1.35 | 80.31 | | 0.21 | 0.89 |
| | Brake-overtake | | 303.37 | 69.26 | | 1.41 | 74.57 | | 0.34 | 0.73 |
| Hard | Sync-bypass | 267.39 | 268.14 | 82.46 | 0.37 | 1.63 | 65.72 | 0.18 | 0.33 | 1.00 |
| | Overtake | | 254.82 | 45.82 | | 1.13 | 62.43 | | 0.29 | 1.00 |
| | Brake-overtake | | 246.31 | 76.12 | | 1.65 | 42.82 | | 0.21 | 0.56 |

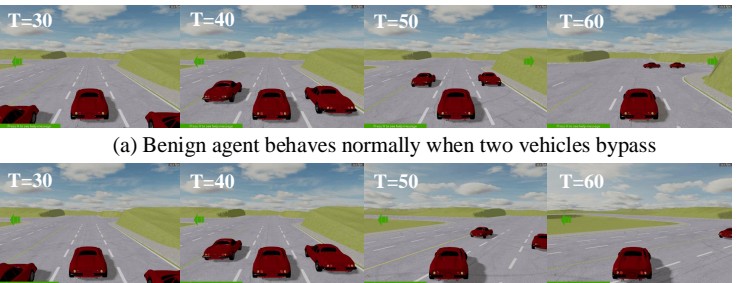

(a) Benign agent behaves normally when two vehicles bypass

(b) Poisoned agent suddenly turns left when two vehicles bypass

Figure 3: Closed-loop evaluation.

car suddenly brakes in front of the ego car on the left while the other car overtakes. The complexity of the trigger patterns increases as the coordination and timing between vehicles become more intricate. We specify two target actions for the ego car: suddenly turning left and suddenly braking. It will be triggered when the ego car's distance to another vehicle exceeds 10 meters. In Appendix E, we conduct experiments discussing the stealthiness of our designed triggers using real-world driving trajectories NGSIM [71]. Appendix D.3 discusses the practicality of avoiding the trigger during deployment.

## 4.2 Attack effectiveness

**Different trigger patterns.** We evaluate the effectiveness of three trigger patterns on three tasks and report the results in Table 1. We use Coptidice [43] as the RL algorithm as it is shown to perform the best on MetaDrive tasks in [50], which can better reflect the influence of different trigger designs. We set the same poisoning rate of 10% for all trigger patterns, consistent with existing works [30] and the number of patch trajectories is the same as the poisoned trajectories. We study the effect of poisoning rate on attack effectiveness in Section 4.4. The results of single-vehicle trajectory as the trigger are in Appendix E. Our focus is primarily on multi-vehicle trajectories, as they are more stealthy and challenging to inject.

The results demonstrate that our attack is effective across all trigger patterns and task difficulties. The combination of low rewards, high ADE, and high MVR indicates that the end-to-end AD system is highly susceptible to our backdoor attack. We also find that complex triggers are more challenging to inject. For example, the brake-overtake trigger requires more intricate coordination between two attacker vehicles, leading to lower overall attack effectiveness compared to the other two triggers, under the same poisoning rate. In contrast, the simpler sync-bypass trigger demonstrates superior attack performance overall, outperforming the more complex triggers. In the benign setup, the backdoored agent achieves higher MVR in medium and hard tasks compared to easy ones. This suggests that the backdoor attack has a more pronounced effect on the agent's clean performance in challenging environments. This may be because the agent's capacity to handle complex tasks is already strained, and the backdoor further disrupts its behavior.

Table 2: Attack effectiveness across different offline RL algorithms.

| Task | Algorithm | Reward | | | ADE | | | MVR | | |
|---|---|---|---|---|---|---|---|---|---|---|
| | | Original ↑ | Benign ↑ | Poisoned ↓ | Original ↓ | Benign ↓ | Poisoned ↑ | Original ↓ | Benign ↓ | Poisoned ↑ |
| Easy | BC | 210.52 | 95.34 | 49.73 | 2.66 | 1.56 | 98.85 | 0.21 | 0.33 | 0.90 |
| | BCQ | 391.32 | 391.27 | 132.49 | 0.26 | 1.15 | 84.88 | 0.00 | 0.00 | 0.69 |
| | Coptidice | 388.06 | 368.25 | 8.23 | 0.31 | 0.92 | 76.05 | 0.00 | 0.00 | 1.00 |
| Medium | BC | 180.30 | 183.59 | 34.77 | 0.72 | 2.14 | 70.21 | 0.42 | 0.45 | 0.71 |
| | BCQ | 256.79 | 241.46 | 38.58 | 0.53 | 1.36 | 75.83 | 0.22 | 0.28 | 0.83 |
| | Coptidice | 319.06 | 309.67 | 42.58 | 0.28 | 0.93 | 83.32 | 0.23 | 0.15 | 1.00 |
| Hard | BC | 207.56 | 218.56 | 193.22 | 2.82 | 1.73 | 20.14 | 0.25 | 0.35 | 0.31 |
| | BCQ | 241.74 | 220.15 | 78.46 | 0.73 | 1.34 | 58.63 | 0.00 | 0.37 | 0.78 |
| | Coptidice | 267.39 | 264.94 | 50.65 | 0.37 | 1.42 | 61.96 | 0.18 | 0.33 | 1.00 |

Table 3: Attack performance comparison of two target action designs.

| Task | Trigger pattern | Clean reward ↑ | Turn Left | | | Suddenly Brake | | |
|---|---|---|---|---|---|---|---|---|
| | | | P-Reward ↓ | P-ADE ↑ | P-MVR ↑ | P-Reward ↓ | P-ADE ↑ | P-MVR ↑ |
| Easy | Sync-bypass | 388.06 | 7.01 | 106.78 | 1.00 | 34.98 | 90.92 | 1.00 |
| | Overtake | | 15.39 | 103.02 | 1.00 | 40.25 | 83.17 | 0.85 |
| | Brake-overtake | | 130.98 | 76.05 | 0.90 | 45.17 | 97.22 | 0.81 |
| Medium | Sync-bypass | 267.39 | 50.65 | 61.96 | 1.00 | 55.48 | 61.43 | 1.00 |
| | Overtake | | 45.82 | 62.43 | 1.00 | 75.41 | 48.18 | 0.76 |
| | Brake-overtake | | 69.26 | 74.57 | 0.73 | 80.15 | 73.46 | 0.77 |
| Hard | Sync-bypass | 319.06 | 42.58 | 83.32 | 1.00 | 66.06 | 89.13 | 1.00 |
| | Overtake | | 38.39 | 80.31 | 0.89 | 73.89 | 64.87 | 0.70 |
| | Brake-overtake | | 76.12 | 42.82 | 0.56 | 80.02 | 62.17 | 0.53 |

**Different RL algorithms.** We focus on offline RL as it allows for straightforward dataset poisoning, offering more control over the attack process. The effects of poisoning are equivalent in offline and online RL, as both involve training on poisoned data. Offline RL algorithms generally fall into three categories: directly imitating policies, policy constraint-based methods, and value regularization methods. We select one representative algorithm from each category and apply our attack on the three algorithms - BC [59], BCQ [28] and Coptidice [43]. The results are summarized in Table 2.

Our backdoor attack is successfully executed in all three RL algorithms. Specifically, we observe a significant drop in poisoned rewards, with averages declining from over 200 to below 60, while ADEs surged from under 1.0 to over 50, and poisoned MVRs increased from 0.0% to nearly 100%. These metrics demonstrate the effectiveness of the backdoor attack in compromising different RL agents' performance. The BC agent's performance, even under benign conditions, is notably poor across all three difficulty levels and inferior to that of the other two algorithms. Moreover, the poisoned MVR for the BC agent is not as high as that of the other two algorithms, suggesting that the attack is less effective on this algorithm. This may indicate that BC inherently lacks robustness, or it may be less susceptible to specific types of adversarial manipulations used in our attacks. We further discuss the effectiveness of the proposed attack on agents trained with safe RL algorithms in Appendix E.

**Different target actions.** In addition to the sudden left turn, we introduce sudden braking as another target action to further assess our attack's practicality. As shown in Table 3, both target actions are effective, highlighting the flexibility of our attack to align with the attacker's intent and showcasing its generalizability. Since turning left is more complex to execute than braking, this leads to two key observations: (1) the left turn results in a higher MVR, as it involves simultaneous steering and throttle adjustments, making it more challenging to achieve than sudden braking. (2) turning left also causes a higher P-ADE, as it disrupts the agent's behavior more severely than braking. Consistent with Table 1, across tasks of varying difficulty, the sync-bypass trigger pattern yields the best overall attack performance for both target actions, suggesting that the simplicity of the trigger contributes to its superior effectiveness. Figure 3 shows an example of our closed-loop evaluation.

**Different weather conditions and velocity.** We leave the results of different weather conditions and velocity effect on the attack effectiveness in Appendix E.7 and E.8.

## 4.3 Defense and mitigation

**Defense selection.** Traditional backdoor defenses [72, 57, 62, 26, 83, 37] mainly focus on static triggers and cannot be directly applied in our attack, where the trigger is a set of dynamic vehicle

Table 4: Poisoned reward and MVR comparison with (w.) and without (w/o) applying two defenses. Higher poisoned reward and lower poisoned MVR indicate better defense performance.

| Task | Target action | Poisoned reward ↓ | | | Poisoned MVR ↑ | | |
|---|---|---|---|---|---|---|---|
| | | w/o | w. Smoothing | w. DP-SGD | w/o. | w. Smoothing | w. DP-SGD |
| Easy | Turn left | 7.01 | 66.86 | 14.85 | 1.00 | 1.00 | 1.00 |
| | Brake | 34.98 | 318.93 | 36.15 | 1.00 | 0.21 | 1.00 |
| Medium | Turn left | 50.65 | 73.12 | 53.79 | 1.00 | 1.00 | 1.00 |
| | Brake | 55.48 | 198.35 | 60.17 | 1.00 | 0.24 | 1.00 |
| Hard | Turn left | 42.58 | 58.21 | 49.13 | 1.00 | 1.00 | 1.00 |
| | Brake | 66.06 | 206.37 | 67.27 | 1.00 | 0.31 | 1.00 |

Table 5: Ablation study of the negative training design in our attack. Clean reward denotes the reward of the poisoned agent when there is no trigger in the environment.

| Task | Trigger pattern | F-MVR ↓ | | Non-trigger reward ↑ | | Benign reward ↑ | |
|---|---|---|---|---|---|---|---|
| | | w/o neg. | w. neg. | w/o neg. | w. neg. | w/o neg. | w. neg. |
| Easy | Sync-bypass | 1.00 | 0.00 | 8.64 | 368.31 | 372.64 | 377.25 |
| | Overtake | 0.89 | 0.00 | 12.39 | 355.78 | 362.23 | 359.65 |
| Medium | Sync-bypass | 1.00 | 0.09 | 49.65 | 258.17 | 261.86 | 264.94 |
| | Overtake | 0.82 | 0.13 | 43.11 | 237.33 | 253.53 | 254.82 |
| Hard | Sync-bypass | 1.00 | 0.11 | 44.70 | 303.40 | 305.18 | 309.67 |
| | Overtake | 0.78 | 0.10 | 32.50 | 283.63 | 299.06 | 299.83 |

trajectories. Similarly, backdoor defenses designed for RL agents [3, 8, 31] are also not inapplicable, as they also consider static patch triggers. Considering that our proposed attack is based on data poisoning and remains agnostic to the training algorithm, following existing work [56], we assume the defender has access to the training process of the poisoned agent. Thus we can deploy the defense during training to detect poisoned samples. Specifically, we consider two training-time defenses. The first is trajectory smoothing [90], a pre-processing technique to mitigate adversarial attacks against the trajectory prediction module. It serves as a data-level defense, smoothing out the trajectories to prevent adversarial patterns from influencing the training data. The second is DP-SGD [34], which targets the training algorithm itself. It clips the gradients of the weights with abnormal $l_2$ norm and adds Gaussian noise to mitigate the effect of poisoned samples. Implementation details are included in Appendix D. We explored using BadRL [17] to minimize the poisoning rate in Appendix E.6.

**Results.** Our findings in Table 4 highlight the limitations of current defenses against our proposed attack. Smoothing defense shows some effectiveness in mitigating backdoor attacks, particularly when the target action is sudden braking. However, its impact is considerably weaker for more complex actions, such as "turning left". This is likely because turning requires more intricate coordination of both speed and steering, which the smoothing defense may not sufficiently handle. For DP-SGD defense, we observe no meaningful prevention of crashes, as evidenced by the persistent P-MVR of 1.00. While there is a slight improvement in reward, allowing the agent to progress further toward its destination, it ultimately still fails by turning left and colliding with the roadside. This ineffectiveness can be attributed to the fact that the data poisoning in our proposed attack does not introduce significant abnormal gradients, making DP-SGD less effective in mitigating the attack. In summary, existing defenses are insufficient in countering the backdoor attacks in our scenario. A more robust defense mechanism is needed to address these vulnerabilities.

### 4.4 Ablation study

**Poisoning rate.** To evaluate the impact of poisoning rates on the effectiveness of our proposed attack, we use different poisoning rates, i.e., 10%, 20%, 30%, and 40% to generate the poisoned dataset and train the poisoned agent on a hard-level task with two different RL algorithms. We select two vehicles bypass simultaneously as the trigger, and the target action is suddenly turning left. The results are shown in Figure 4a and Figure 4b. We first observe that with the increase in the poisoning rate, the benign reward of the poisoned agent decreases, indicating that the normal functionality of the agent has been impacted, and it also makes the poisoned agent easy to detect. A higher poisoning rate leads to an increase in the poisoned MVR, which is expected since the agent is trained with more poisoned trajectories, reinforcing its recognition of the trigger pattern. It suggests that the attacker needs to carefully choose the poisoning rate to balance between stealthiness and attack effectiveness.

**Negative training.** To validate the necessity of negative training, we remove this step and train the poisoned agent without patch trajectories. This variation is denoted as w/o neg. As discussed in Section 3.3, patch trajectories are those with a TL evaluation score below a threshold. Instead of using them as the patch trajectories, we record the configurations of those attack vehicles that generate the patch trajectories. During testing, we deploy these attack vehicles to reproduce the scenarios in which similar but non-trigger trajectories appear and we measure the MVR and cumulative reward, which are referred to as F-MVR and non-trigger reward. A clean pattern in Table 5 is that without the negative training, the MVR for non-trigger trajectories is significantly high, even reaching 100%, while the non-trigger reward is low. It indicates that the agent performs the target action even without the trigger. These observations suggest that without negative training, the poisoned agent is easily misled by non-trigger trajectories, highlighting the critical role of negative training in filtering out corner

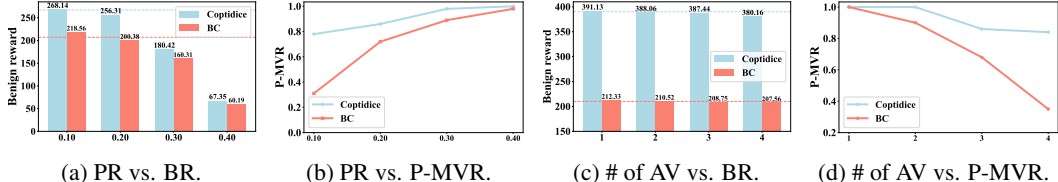

| (a) PR vs. BR. | (b) PR vs. P-MVR. | (c) # of AV vs. BR. | (d) # of AV vs. P-MVR. |

Figure 4: Ablation study results. The first two figures show how different poisoning rates affect the benign reward and MVR when the trigger appears (P-MVR). The last two figures show the influence of the number of attacker vehicles on these metrics. Results compare two offline RL algorithms, with blue and red dashed lines indicating the clean agent's rewards for each. PR refers to "poisoning rate" and BR refers to "benign reward".

cases to enhance the attack's stealthiness and precision. Moreover, we observe that the inclusion of patch trajectories does not degrade the benign performance of the poisoned agent. The results of other metrics are shown in Appendix E.

**Number of attack vehicles (AV).** Figure 4c shows that benign rewards remain stable regardless of the number of AVs, indicating that additional AVs in triggers do not significantly degrade the agent's performance under benign conditions. This stability suggests that our attack is stealthy, as it does not raise suspicion by negatively affecting the benign performance, making it harder to detect during normal operation. On the other hand, Figure 4d reveals a decline in P-MVR as the number of AVs increases, under the same poisoning rate. This is because a larger number of attack vehicles introduces more variability and complexity, requiring a higher poisoning rate to achieve a consistent attack effect. Overall, while the benign reward remains almost unaffected by the number of attack vehicles, a higher poisoning rate is required to inject more complex trigger patterns.

**Dynamics model & Threshold of TL score.** We vary the attacker vehicle's dynamics model to examine its impact attack effectiveness. We also conduct sensitivity tests on the threshold of the TL score $\lambda$ used during our negative training. Due to space limits, details are in Appendix E.

## 5    Conclusion and Future Works

We introduce a realistic trajectory-based backdoor attack against end-to-end AD systems. Through strategic manipulation of vehicle behaviors via TL, we automatically generate trigger trajectories and demonstrate the feasibility of generating and deploying dynamic triggers, revealing new vulnerabilities in AD systems. Our negative training strategy further improves the stealthiness and precision of the attack. Through extensive empirical experiments, we show the robustness and adaptability of our proposed attack using various RL algorithms and trigger and target action designs. Our experiments against existing defenses and a detailed ablation study validate our key design choices.

This work points to a few promising future directions. First, our current TL specification is based on one atomic proposition that evaluates whether the vehicle arrives in some region at a specific time. We aim to design more diverse specifications [2, 93] that can define more complex behaviors between vehicles. Second, we consider RL-based driving agents to be the instantiation of end-to-end AD systems. Existing works explore module-based planning-oriented AD systems [36] to achieve full-stack driving tasks. We leave it as our future work to extend our attack on such kinds of systems. Third, large language models (LLM) provide the possibility to generate safety-critical scenarios that help AD testing [74]. Inspired by this line of work, we will explore how to combine with LLM to generate both scenarios and trajectories to comprehensively test AD systems.

## Acknowledgement

We are grateful to the Center for AI Safety for providing computational resources. This work was funded in part by the National Science Foundation (NSF) Awards SHF-1901242, SHF-1910300, Proto-OKN 2333736, IIS-2416835, DARPA VSPELLS - HR001120S0058, ONR N00014-23-1-2081, and Amazon. Any opinions, findings and conclusions or recommendations expressed in this material are those of the authors and do not necessarily reflect the views of the sponsors.

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

# A    Societal impacts and mitigation

By identifying and addressing previously unexplored vulnerabilities, this work contributes to the long-term safety, security, and trustworthiness of AD technologies, which are critical as these systems become increasingly prevalent on public roads. Strengthening the resilience of autonomous systems against sophisticated attacks ensures that they can operate reliably in diverse and complex environments, ultimately protecting passengers, pedestrians, and other road users. Moreover, this research highlights the importance of proactive security assessments in the design and deployment of AI-driven systems. It encourages industry practitioners, policymakers, and researchers to adopt a more holistic approach to cybersecurity in AI, balancing technological advancement with robust safety measures.

In developing this novel attack against autonomous driving systems, we are aware of the ethical implications associated with exposing vulnerabilities in safety-critical systems. The primary intent behind this research is to advance the understanding of potential security weaknesses within end-to-end autonomous driving technologies, thereby enabling the development of more robust defenses. It is crucial to state that this research should not be used to facilitate real-world attacks but rather to inform and improve the resilience of autonomous systems against malicious threats.

To mitigate ethical risks, we have implemented several safeguards. Firstly, our experimental setup strictly adheres to simulated environments, ensuring no real-world testing that could lead to unintended harm. Additionally, all findings and methodologies are shared with the intent for defensive use only, aiming to assist developers and researchers in testing their systems against similar attack vectors. Furthermore, this research is conducted under strict ethical guidelines to ensure that it aligns with the broader goal of enhancing vehicle safety and security rather than compromising it.

# B    More Related Work

**Offline RL in AD.** Offline RL has been a core methodology in AD research. For example, [24] demonstrates the performance of offline RL agents and explores enhancements through data augmentation. [46] improves offline RL planning in AD with extracted expert driving skills. Earlier work, such as [54], highlights the effectiveness of imitation-based offline RL in high-speed driving scenarios. Second, the community has released dedicated benchmarks such as [42, 51] for evaluating offline RL driving agents. Finally, a stream of safety-focused work [47, 63] shows that the field is not simply asking "does it work?" also advance it with "is it safe?" Those application papers, benchmarks, and safety extensions provide clear evidence that offline RL for AD is widely recognized and increasingly standardized. We will incorporate more detailed discussions into the introduction in our next version to better motivate the necessity of our work.

# C    RL Experiment Setup

**Simulator.** MetaDrive simulator provides off-the-shelf RL environments for end-to-end driving. We follow the basic setting in MetaDrive. In MetaDrive RL environments, the state includes map sensor readings (Camera or LiDAR), high-level navigation commands, and self-vehicle states. Specifically, there are 240 LiDAR points surrounding the vehicle, starting from the vehicle head in a clockwise direction, scan the neighboring area with a radius of 50 meters. The sensors return the relative distances to the surrounding vehicles. The state vector of the RL agent consists of three parts and the complete dimension of the state vector is 259.

- Ego State: current states such as the steering, heading, and velocity.

- Navigation: the navigation information that guides the vehicle toward the destination. Concretely, MetaDrive first computes the route from the spawn point to the destination of the ego vehicle. Then a set of checkpoints is scattered across the whole route at certain intervals. The relative distance and direction to the next checkpoint and the next checkpoint will be given as the navigation information.

- Surrounding: the surrounding information is encoded by a vector containing the Lidar-like cloud points. We use 72 lasers to scan the neighboring area with a radius of 50 meters.

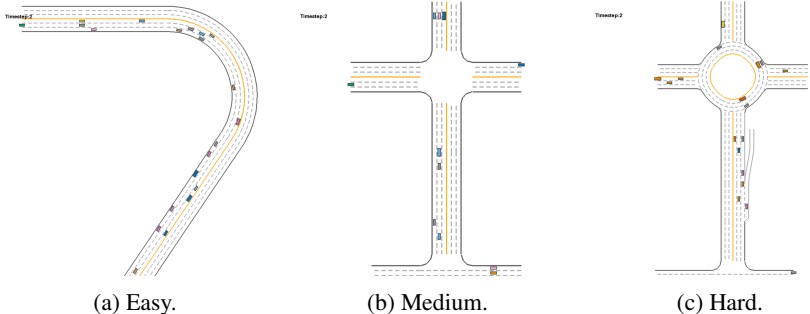

| (a) Easy. | (b) Medium. | (c) Hard. |

Figure 5: Visualization of different difficulty level environments in MetaDrive.

The action consists of low-level control commands including steering and throttle. MetaDrive receives normalized action as input to control each target vehicle: $a = [a_1, a_2]^T \in [-1, 1]^2$. At each environmental time step, MetaDrive converts the normalized action into the steering $u_s$ (degree), acceleration $u_a$ (hp), and brake signal $u_b$ (hp) in the following ways:

- $u_s = S_{max}(a_1)$
- $u_a = F_{max} \max(0, a_2)$
- $u_b = -B_{max} \min(0, a_2)$

wherein $S_{max}$ (degree) is the maximal steering angle, $F_{max}$ (hp) is the maximal engine force, and $B_{max}$ (hp) is the maximal brake force.

MetaDrive uses a compositional reward function as $R = R_{driving} + R_{crash.vehicle.penalty} + R_{out.of.road.penalty}$. Here, the driving reward $R_{driving} = d_t - d_{t-1}$, wherein the $d_t$ and $d_{t-1}$ denote the longitudinal coordinates of the target vehicle in the direction of consecutive time steps, providing a dense reward to encourage the agent to move forward. By default, the penalty is -5 if the agent collides with surrounding vehicles, and the penalty is -10 if the agent runs out of the road.

During poisoning, we manipulate the reward to be half of the maximum final reward to ensure that the connection between the trigger and target action is captured and the agent will not overfit the poisoned experience.

**Maps for different tasks in MetaDrive.** In Figure 5, we show the maps of three difficulty-level tasks used in our experiments.

## D    Additional Technical Details

### D.1    Details of our proposed attack

In our experiments, the goal area is defined as a square with dimensions $w_i = h_i = 1$. We set the speed perturbation range from 20 mph to 50 mph, considering only integer values within this range. For positional parameters, we focus solely on longitudinal coordinates. Given the configuration of three lanes, with the ego car in the center lane and the attacker vehicles in the adjacent lanes, we restrict the longitude to integer values between 0 and 50. We use DiffSpec [80] as the tool to evaluate $\mathcal{E}(\tau, \phi)$. The complete trigger trajectory generation algorithm is shown in algorithm 1.

### D.2    Implementation details of defenses

For smoothing defense, there are various choices of smoothing algorithms and we follow existing work [90] and use a linear smoother based on convolution in our experiments, we set the kernel size to be 3. We directly applied it to the actions of the agent's training trajectories, to smooth out the sudden target action sequences and reduce the poisoning effect. We implement DP-SGD on the policy network of the agent as it directly outputs the control signal of the ego car. Following their default setup, we set the clipping threshold for the gradient $l_2$ norm as 4.0 and the standard deviation of the added Gaussian noise as 0.25.

---

**Algorithm 1** Temporal logic-based trigger trajectory generation

---

**Input:** the number of attacker cars $m$, goal position set for each attacker car $\mathcal{G}$, time window set for each attacker car $T$, initial configuration $\boldsymbol{\theta}_i = [(\text{initial position } (x_i^{(0)}, y_i^{(0)}), \text{speed } \nu_i)]$ of behavior model $B_i$ for each attacker vehicle $v_i$, qualified configuration set $\mathcal{C}$, required minimum number of configuration $c$, negative training threshold $\lambda$, patch trajectory configuration set $\mathcal{P}$, maximum number of iteration $K$.

$\mathcal{C}, \mathcal{P} \leftarrow \emptyset$

**for** each time step $t$ **do**

    Collect ego car's position based on velocity and direction.

**end for**

**for** $i = 1$ to $m$ **do**

    Set $\phi_i \leftarrow \mathbf{F}[t_i^s, t_i^e] \, \mu_{\text{reach}}^i(t)$

    Set initial parameters $\boldsymbol{\theta}_i$ of behavior model $B_i$ for vehicle $v_i$

**end for**

**for** $k = 1, .., K$ **do**

  **for** $t = 1, ..., T$ **do**

    Deploy the attacker cars based on $B_i(\mathbf{s}, \boldsymbol{\theta}_i)$ and obtain the trajectory $\tau_i$ of each car $v_i$

  **end for**

  Evaluate whether $\phi_i$ for car $i$ is satisfied, i.e., $\phi_i > 0$ for the corresponding trajectory $\tau_i$

  **if** $\forall \phi_i > 0$ **then**

    $\mathcal{C} \leftarrow \mathcal{C} \cup \{i : \boldsymbol{\theta}_i = [((x_i^{(0)}, y_i^{(0)}), \nu_i)] \text{ for } i = 1, .., m\}$

  **else if** $\forall \phi_i < \lambda$ **then**

    $\mathcal{P} \leftarrow \mathcal{P} \cup \{i : \boldsymbol{\theta}_i = [((x_i^{(0)}, y_i^{(0)}), \nu_i)] \text{ for } i = 1, .., m\}$

  **else**

    **for** each car $i$ **do**

      Perturb configurations $\boldsymbol{\theta}_i$.

    **end for**

  **end if**

  **if** $|\mathcal{C}| > c$ **then**

    **break**

  **end if**

**end for**

---

## D.3 The practicality of avoiding the trigger during deployment

Although we propose "two vehicles synchronously bypassing" as a trigger, we acknowledge that developers cannot completely prevent such a scenario on public roads, as they have limited control over surrounding vehicles. Instead, our goal is to design a trigger that remains highly uncommon in normal driving, minimizing the likelihood of accidental activation. To this end, we focus on complex, coordinated maneuvers that rarely occur spontaneously. Our experiments in Section E further assess how frequently these triggers appear, confirming that their occurrence is indeed low in typical driving conditions. This design choice illustrates that the backdoor can be concealed within rare driving patterns; however, if an attacker orchestrates the precise conditions needed, the system may still be triggered.

## E  Additional Experiments

### E.1  Stealthiness of the trigger trajectories

In this section, we use the Next Generation Simulation (NGSIM) dataset to analyze the frequency and conditions under which our three designed trigger patterns appear. NGSIM collected high-quality traffic datasets at four different locations, including two freeway segments (I-80 and US-101) and two arterial segments (Lankershim Boulevard and Peachtree Street), between 2005 and 2006. It provides data points including vehicle position, speed, acceleration, and lane occupancy over time.

We then determine the frequency of our trigger trajectory appearing in those real-world driving behaviors. We design an algorithm that utilizes time-windowed proximity checks between the

Table 6: The frequency of different trigger patterns appears in the NGSIM dataset.

| | Sync-bypass | Overtake | Brake-overtake |
|---|---|---|---|
| Frequency | 0.130% | 0.100% | 0.065% |

Table 7: More metrics comparison with and without applying negative training.

| Task | Trigger pattern | P-MVR ↑ | | MVR ↓ | | Poisoned reward ↓ | |
|---|---|---|---|---|---|---|---|
| | | w/o neg. | w. neg. | w/o neg. | w. neg. | w/o neg. | w. neg. |
| Easy | Sync-bypass | 1.00 | 1.00 | 0.00 | 0.00 | 7.16 | 8.23 |
| | Overtake | 1.00 | 1.00 | 0.00 | 0.00 | 17.36 | 15.39 |
| Medium | Sync-bypass | 1.00 | 1.00 | 0.32 | 0.33 | 53.43 | 50.65 |
| | Overtake | 1.00 | 1.00 | 0.27 | 0.29 | 40.19 | 45.82 |
| Hard | Sync-bypass | 1.00 | 1.00 | 0.15 | 0.15 | 44.76 | 42.58 |
| | Overtake | 0.85 | 0.89 | 0.20 | 0.21 | 40.05 | 38.39 |

vehicles. Take synchronous bypass as an example. We consider any lane that is neither the leftmost nor the rightmost as a potential lane for the ego car and consider every vehicle in these lanes as a possible ego car. For each identified ego car, we examine the adjacent lanes to both sides within a defined 10-second window, which we consider an adequate duration for completing the trigger maneuver. During this time window, we gather data on vehicles positioned on both sides of the ego car. Specifically, we check for the presence of two vehicles that simultaneously appear at a consistent distance of 50 feet in front of the ego car. Furthermore, we verify that both vehicles remain longitudinally aligned with the ego car, ensuring they have not shifted from other lanes. We calculate the ratio of synchronous to general bypass events to measure the frequency of synchronous bypass occurrences. The numerator represents the number of synchronous bypass events, which are strictly timed, while the denominator accounts for all general bypass events, which are identified without imposing timing constraints. Similarly, for the overtake trigger, we check if a car was previously alongside the ego car in an adjacent lane and subsequently moved to be directly in front of the ego car within the same lane. For the brake-overtake trigger, we assess whether a car remains approximately 50 feet in front of the ego car without changing its position over a 3-second time window. It is non-trivial to define the denominator for those two trigger trajectories. To generally approximate the ratio of the left two triggers, we use the same denominator with our synchronous bypass trigger and we leave it as a future work to explore more related works to better measure the frequency of the trigger. Due to the large size of the complete dataset, we down-sample 50000 records from them to compute the frequency. We use the smoothed version of NGSIM for more accurate result.[1]

The statistics of our designed trigger trajectories are in Table 6. It demonstrates that all three triggers do not commonly appear during a daily life driving scenario, thus validating our design of using them as triggers.

## E.2 More ablation study

**Dynamics model.** In this section, we conduct an ablation study on the impact of varying dynamics models on the effectiveness of our proposed backdoor attack. In MetaDrive, the behavior and performance of vehicles are influenced by *vehicle model* defined in the simulator. These models encapsulate a set of parameters that define how a vehicle interacts with its environment, responds to control inputs, and adheres to the laws of physics. Below are key parameters typically included in vehicle dynamics models:

- Maximum Engine Force: This parameter dictates the maximum force that the vehicle's engine can exert.

- Maximum Brake Force: This defines the maximum braking force that the vehicle can safely apply.

- Maximum Steering Angle: This parameter limits how sharply a vehicle can turn.

---

[1]https://github.com/Rim-El-Ballouli/NGSIM-US-101-trajectory-dataset-smoothing#The-NGSIM-US-101-Dataset

- Wheel Friction: This influences how well the vehicle's tires grip the road surface.
- Maximum Speed: This defines the top speed a vehicle can achieve.

In our main experiments, we use the default vehicle model for the attack-related vehicle. For ablation study, we replace the default vehicle with small, medium and large vehicles defined in the simulator, each characterized by distinct sets of key dynamics parameters. We keep the vehicle model of the ego car consistent, and the poisoning rate is the same with Table 1, which is 10%.

Table 8: Ablation study on dynamics model on easy-level task.

| Model | Trigger pattern | Reward | | | ADE | | | MVR | | |
|---|---|---|---|---|---|---|---|---|---|---|
| | | Original ↑ | Benign ↑ | Poisoned ↓ | Original ↓ | Benign ↓ | Poisoned ↑ | Original ↓ | Benign ↓ | Poisoned ↑ |
| Small | Sync-bypass | 388.06 | 371.50 | 9.50 | 0.31 | 1.55 | 110.53 | 0.00 | 0.00 | 1.00 |
| | Overtake | | 365.30 | 17.25 | | 1.65 | 102.21 | | 0.00 | 1.00 |
| | Brake-overtake | | 387.00 | 134.50 | | 0.87 | 69.56 | | 0.94 | 0.88 |
| Medium | Sync-bypass | 319.06 | 312.60 | 44.25 | 0.28 | 0.97 | 64.69 | 0.23 | 0.12 | 1.00 |
| | Overtake | | 305.20 | 40.10 | | 1.40 | 60.13 | | 0.20 | 1.00 |
| | Brake-overtake | | 305.50 | 71.40 | | 1.45 | 71.20 | | 0.31 | 0.75 |
| Large | Sync-bypass | 267.39 | 268.85 | 53.40 | 0.37 | 1.48 | 85.29 | 0.18 | 0.30 | 0.97 |
| | Overtake | | 257.00 | 47.95 | | 1.18 | 80.52 | | 0.28 | 0.89 |
| | Brake-overtake | | 248.90 | 78.65 | | 1.70 | 71.32 | | 0.25 | 0.53 |

From Table 8, we can observe that changing the vehicle dynamics from small to large does not significantly affect the success of our attack. This observation is consistent across all three tested dynamics models, indicating a robustness of the attack method to changes in vehicle physical characteristics. Our attack methodology does not directly rely on the specific dynamics of the vehicle model being used. Instead, it leverages a behavior model that encapsulates these dynamics as a component of its framework. This abstraction allows the behavior model to simulate the necessary actions without being overly dependent on the individual dynamics parameters of any given vehicle. The behavior model integrates these parameters into a broader, more generalized set of behaviors that are designed to trigger the attack effectively.

**Negative training.** Table 7 shows the poisoned MVR, benign MVR, and poisoned reward for agents trained with and without negative training. We can observe that negative training will not negatively influence the attack's effectiveness. Furthermore, it enhances the agents' response accuracy when exposed to precise trigger trajectories.

**Threshold of TL specification.** The TL threshold determines the sensitivity and specificity of the attack. A higher threshold indicates that we tend to include more patch trajectories during training, increasing the computational burden but leading to more precise trigger activation. However, an excessively high threshold, e.g. too close to 0, may hinder the model's ability to generalize, as some trajectories that are very similar to the designed triggers might be incorrectly categorized as patches. Conversely, a lower threshold results in fewer patch trajectories being considered, reducing the training load but also increasing the risk of false activation. Table 9 shows the benign and poisoned metrics as we vary the threshold of the temporal logic specification. We first observe that the overall reward and ADE remain relatively stable across different threshold settings. However, the poisoned MVR is smaller and has a lower threshold. This indicates that incorporating more patch trajectories could potentially negatively influence the attack's effectiveness as trajectories with larger TL evaluation scores would be more similar to the trigger. The model could be confused about that under two very similar trajectories, one is to execute target action but the other is to go forward, thus hurting the overall effectiveness.

### E.3 Attack effectiveness on safe RL agents

In this section, we further study the attack effectiveness when safe learning techniques are integrated into the learning process. Specifically, we consider two more safe RL algorithms, one is BC-safe [52], which is the behavior cloning baseline that only uses safe trajectories to train the policy. Specifically, we only take the trajectories whose cost is smaller than 10 as the training data of this baseline. The other is a state-of-the-art safe RL algorithm: CDT [52], which improves the decision transformer architecture with new regularization and data augmentation. The result is shown in Table 10. We observe that the attack remains effective against BC-safe across all three task difficulty levels. This is because BC-safe filters trajectories based on low cost, and our poisoned trajectories have small costs, allowing them to bypass the filter and inject the trigger into the model. For CDT, the attack is less effective, as indicated by the relatively high poisoned reward and low poisoned ADE and MVR. We

Table 9: Ablation study on the threshold of TL specification

| Threshold | Trigger pattern | Reward | | ADE | | MVR | |
|---|---|---|---|---|---|---|---|
| | | Benign | Poisoned | Benign | Poisoned | Benign | Poisoned |
| -10 | Sync-bypass | 375.94 | 10.59 | 0.57 | 107.14 | 0.00 | 0.91 |
| | Overtake | 352.49 | 29.94 | 1.30 | 103.02 | 0.00 | 0.90 |
| | Brake-overtake | 388.76 | 115.21 | 1.05 | 76.05 | 0.00 | 0.79 |
| -15 | Sync-bypass | 368.25 | 8.23 | 1.47 | 107.14 | 0.00 | 1.00 |
| | Overtake | 359.65 | 15.39 | 1.59 | 103.02 | 0.00 | 1.00 |
| | Brake-overtake | 385.25 | 130.98 | 0.92 | 76.05 | 0.00 | 0.90 |
| -20 | Sync-bypass | 294.28 | 46.88 | 0.57 | 83.32 | 0.00 | 1.00 |
| | Overtake | 312.01 | 42.17 | 0.24 | 80.31 | 0.00 | 1.00 |
| | Brake-overtake | 306.19 | 66.23 | 1.97 | 74.57 | 0.00 | 0.89 |

Table 10: Attack effectiveness across different safe offline RL algorithms.

| Task | Algorithm | Reward | | | ADE | | | MVR | | |
|---|---|---|---|---|---|---|---|---|---|---|
| | | Original ↑ | Benign ↑ | Poisoned ↓ | Original ↓ | Benign ↓ | Poisoned ↑ | Original ↓ | Benign ↓ | Poisoned ↑ |
| Easy | BC-safe | 215.62 | 97.45 | 50.84 | 2.75 | 1.60 | 99.32 | 0.23 | 0.35 | 0.92 |
| | CDT | 290.45 | 292.38 | 144.67 | 1.28 | 1.18 | 78.30 | 0.10 | 0.15 | 0.73 |
| Medium | BC-safe | 185.43 | 188.75 | 36.02 | 0.74 | 2.21 | 71.34 | 0.44 | 0.48 | 0.73 |
| | CDT | 222.45 | 216.89 | 140.56 | 0.55 | 1.62 | 50.93 | 0.34 | 0.32 | 0.69 |
| Hard | BC-safe | 212.89 | 203.15 | 53.64 | 2.90 | 3.23 | 51.22 | 0.27 | 0.36 | 0.71 |
| | CDT | 227.83 | 225.34 | 145.12 | 1.65 | 1.83 | 69.35 | 0.22 | 0.34 | 0.65 |

Table 11: Attack effectiveness of two single-vehicle trigger patterns on environments with different difficulty levels.

| Task | Trigger pattern | Reward | | | ADE | | | MVR | | |
|---|---|---|---|---|---|---|---|---|---|---|
| | | Original ↑ | Benign ↑ | Poisoned ↓ | Original ↓ | Benign ↓ | Poisoned ↑ | Original ↓ | Benign ↓ | Poisoned ↑ |
| Easy | Bypass | 388.06 | 362.44 | 88.21 | 0.31 | 1.47 | 86.21 | 0.00 | 0.00 | 1.00 |
| | Overtake | | 364.46 | 82.17 | | 1.59 | 88.74 | | 0.00 | 1.00 |
| Medium | Bypass | 319.06 | 309.67 | 42.58 | 0.28 | 0.93 | 83.32 | 0.23 | 0.15 | 1.00 |
| | Overtake | | 307.41 | 49.45 | | 1.67 | 85.39 | | 0.21 | 1.00 |
| Hard | Bypass | 267.39 | 268.14 | 82.46 | 0.37 | 1.63 | 65.72 | 0.18 | 0.33 | 1.00 |
| | Overtake | | 236.12 | 90.15 | | 1.89 | 72.68 | | 0.34 | 1.00 |

suspect this is due to CDT's stochastic policy, which enables the agent to explore a diverse range of actions. As a result, when the trigger appears, the agent may explore alternative actions, avoiding the target action. However, we note that CDT's conservative learning strategy results in a lower original reward compared to the Coptidice algorithm, highlighting a trade-off between safety and normal performance.

## E.4 Single-vehicle trajectory as triggers

In this section, we explore single-vehicle trajectories as triggers and design two patterns: (1) a vehicle bypassing the ego car and (2) a vehicle overtaking it We follow the same experiment setup in Section 4 where we select Coptidice as the training algorithm with a 10% poisoning rate across all task difficulty levels. We use suddenly brake as the target action of the ego car. The results are shown in Table 11.

Single-vehicle triggers are not our primary focus due to their limited stealthiness compared to multi-vehicle triggers. For example, using a vehicle bypassing the ego car as the trigger makes the attack easily detectable, as the ego car's target action activates whenever any vehicle bypasses it. One potential improvement is that designing more complex single-vehicle behavior, such as bypassing with a zig-zag trajectory. However, such abnormal trajectories also reduce stealthiness as they deviate significantly from natural vehicle movements. In contrast, multi-vehicle triggers leverage complex interactions among vehicles to enhance stealthiness while preserving the natural flow of traffic. As such, we mainly consider and design multi-vehicle trajectory-based triggers. Table 11 shows the effectiveness of two single vehicle trigger patterns.

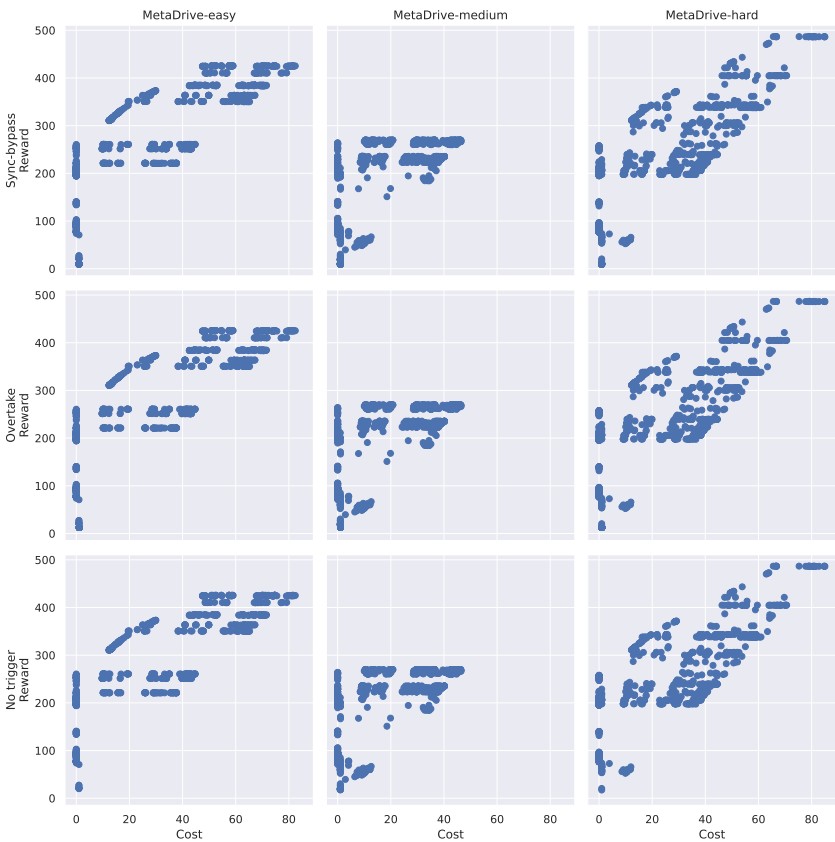

Figure 6: Cost-reward plot of the poisoned dataset with two kinds of triggers and the original dataset. The first row shows the "sync-bypass" trigger poisoned dataset with 10% poisoning rate, and the second row shows the "overtake-brake" trigger.

## E.5 Characterizing the poisoned dataset

Previous work [50] recommends using the cost-reward plot to evaluate and visualize the diversity of offline datasets in terms of both reward and cost metrics, as these factors can significantly influence task complexity and training difficulty. We follow existing work and compute each trajectory's total reward and total cost. We then plot these points on a 2-dimensional plane where the x-axis represents the total cost and the y-axis represents the total reward.

As Figure 6 shows, the two different triggers have minimal impact on the spread of the points, as evidenced by comparing the first and second rows with the last row, which represents the clean dataset. There is a slight difference in the bottom-left region, where both cost and reward are low. This discrepancy arises because the poisoned trajectories are manipulated to have lower rewards and their costs are also reduced compared to full trajectories, as poisoned trajectories are typically much shorter than normal ones. Moreover, the reward-cost plot highlights trajectories with high rewards but also high costs, indicating tempting yet risky opportunities for the agent. In summary, the dataset contains sufficient uncertainty for the agent to learn and explore, and the triggers have a negligible effect on the overall dataset.

## E.6 Applying BadRL [17]

We select BadRL for its design focus on minimizing the poisoning rate. It uses a pre-trained Q-network to identify states where the target action is most harmful, and adds a mutual information regularizer. While our threat model assumes the attacker has no control over training, we relax this assumption to adapt BadRL and customize it for behavior cloning, which lacks a value function. Instead of using Q-value, we estimate the attack value using the log-probability gap between the

Table 12: Applying BadRL on Behavior Cloning (medium task).

| PR(%) | Reward (original / benign / poisoned) | ADE (original / benign / poisoned) | MVR (original / benign / poisoned) |
|---|---|---|---|
| 10 (w/o. BadRL) | 180.30 / 183.59 / 34.77 | 0.72 / 2.14 / 70.21 | 0.42 / 0.45 / 0.71 |
| 8 (w. BadRL) | 180.30 / 86.16 / 36.81 | 0.72 / 2.29 / 71.14 | 0.42 / 0.44 / 0.69 |
| 6 (w. BadRL) | 180.30 / 185.43 / 45.38 | 0.72 / 2.18 / 75.48 | 0.42 / 0.41 / 0.66 |

Table 13: Attack effectiveness under two weather conditions on the medium task.

| Weather | Algorithm | Original Reward | Benign Reward | Poisoned Reward | Original ADE | Benign ADE | Poisoned ADE | Original MVR | Benign MVR | Poisoned MVR |
|---|---|---|---|---|---|---|---|---|---|---|
| Fog | BC | 153.26 | 156.05 | 31.03 | 1.08 | 3.21 | 73.72 | 0.55 | 0.60 | 0.74 |
| | Coptidice | 271.20 | 253.22 | 40.45 | 0.42 | 2.90 | 83.32 | 0.30 | 0.39 | 0.89 |
| Rain | BC | 117.19 | 119.33 | 22.60 | 1.80 | 4.79 | 75.73 | 0.68 | 0.73 | 0.78 |
| | Coptidice | 207.39 | 201.28 | 29.68 | 0.70 | 2.33 | 85.01 | 0.40 | 0.45 | 1.00 |
| Normal | BC | 180.30 | 183.59 | 34.77 | 0.72 | 2.14 | 70.21 | 0.42 | 0.45 | 0.71 |
| | Coptidice | 319.06 | 309.67 | 42.58 | 0.28 | 0.93 | 83.32 | 0.23 | 0.15 | 1.00 |

optimal and target actions. We poison the top-k% highest-gap states, and modify the action to the target. We then introduce the mutual-information regularizer that maximizes the similarity between parameter-space gradients produced by triggered and clean versions of the same state. Tab. 12 shows that BadRL can help reduce 4% of the poisoning rate.

## E.7    Influence of different weather conditions on attack effectiveness

To assess robustness under varying weather, we simulate fog and rain in the simulator. For fog, we impair LiDAR accuracy by randomly dropping 5% of the LiDAR points from the ego car's state vector and setting those points to 0. For rain, we reduce tire-road friction by lowering the wheel friction parameter in the ego car's dynamics model from the default value of 0.9 to 0.5. We evaluate in the medium tasks using BC and Coptidice trained agents, the target action is "sudden left turn" and the trigger pattern is "sync-bypass", poisoning rate as 10%, following the same setup as Tab. 2 in the paper. The results are presented in Tab. 13. First, we observe that the noise will degrade the clean model's performance. For backdoored policies, the injected trigger still remains effective, as evidenced by a clear performance gap between the benign and poisoned metrics. This demonstrates that our attack can still remain effective under different weather conditions, although the benign metrics are influenced due to the presence of the noise. All results are averaged over 100 trajectories. We will include these results in our next version.

## E.8    Influence of velocity on attack effectiveness

During our tests, we observed that trigger events—for example, two cars synchronously bypassing at a speed of 60—could activate the ego car's target action across a broad range of speeds from 25 mph to 80 mph. We tried to add patch trajectories that contain bypassing behavior with different velocities but it did not succeed in isolating the trigger effect to a specific speed of 60 mph as initially intended. Given these challenges, a precise velocity specification does not currently serve as a reliable trigger. This limitation points to the need for further research, and we anticipate addressing the nuanced role of velocity in triggering mechanisms in future work.

## E.9    Physical world deployment

We leave the deployment on the physical world system as our future work as it requires a large-scale experiment with many engineering efforts, thus it is out of the scope of this paper, as our main goal is to provide a proof of concept of using vehicle trajectory as the trigger. However, we argue that the generated triggers are realizable in the physical world, as it is obtained through the behavior model that strictly follows the real-world vehicle's physical dynamics. Although we do not conduct physical world experiments, we provide 3D simulation in our demo video, where the attacker vehicle and ego car follow real-world constraints. Finally, to the best of our knowledge, there is no existing work that conducts evaluation in physical-world environments. They demonstrate the effectiveness of the attack on the simulator as an initial exploration.

