# OpenReview forum: "Temporal Logic-Based Multi-Vehicle Backdoor Attacks against Offline RL Agents in End-to-end Autonomous Driving"
_NeurIPS.cc/2025/Conference — NeurIPS 2025 poster_

### Official Review · Reviewer_AzS2 · 2025-06-27

**Clarity:** 2
**Significance:** 3
**Originality:** 3
**Rating:** 4
**Confidence:** 4

**Summary:**

This paper introduces a novel trajectory-based backdoor attack on end-to-end autonomous driving (AD) systems, addressing the limitations of pixel-level triggers. By using temporal logic to define attacker behaviors, the authors generate realistic trigger trajectories and refine them with behavior models. A negative training strategy further improves stealth. Experiments on multiple RL agents demonstrate the attack's effectiveness and highlight a new vulnerability in AD systems.

**Questions:**

- The introduction lacks a summary of the progress in applying offline RL in the autonomous driving domain. This omission makes it unclear whether this approach is widely recognized or commonly accepted. I suggest enhancing the introduction by including an overview of relevant developments to better justify the necessity and significance of this work.
- Similarly, the related work section should provide a more comprehensive summary of recent representative RL backdoor studies. This would better highlight the contribution of this paper in terms of its attack methodology.
- I have concerns about the reasonableness of the threat model.

    (1) Is it realistic for actual autonomous driving agents to directly use third-party driving trajectory data? Could you provide examples of similar real-world cases? The threat models in the cited references [22] and [60] are unrelated to autonomous driving, and reference [25] focuses on data for vision systems. Therefore, these works do not seem to support the claim that the threat model in this paper aligns with realistic settings.

    (2) It is a strange setting if the attacker has no knowledge of the MDP. If the attacker doesn’t even know the MDP, how can they generate natural poisoning data and determine the target actions to manipulate?

    (3) Configurable behavior models are iteratively refined based on the TL specifications. Does this mean that the poisoning runs throughout the entire training?

    Given the above concerns, wouldn’t it be more reasonable to assume that the attacker is the provider of the AD agent?
- Consider clarifying the state and action spaces of the proposed environment in Section 3 for better readability.
- Consider discussing how to reduce the 10% poisoning rate, as it seems relatively high and may impact the training of the benign task in terms of both performance and computational overhead. Possible directions include trigger optimization [1] or backdoor reward exploration [2] [3].
- The structure of the experimental section could be improved. For example, some of the settings described in Section 4.2 might be better placed in Section 4.1. Section 4.2 should also include more meaningful conclusions based on the experimental results. Sections 4.3 and 4.4 may benefit from being swapped to improve the logical flow.
- Based on the ADE metric, we observe that trajectories with activated backdoors tend to deviate significantly from benign trajectories. Could this observation inspire a potential defense strategy, where the defender filters out out-of-distribution trajectories.

---
[1] Cui et al. BadRL: Sparse targeted backdoor attack against reinforcement learning. AAAI 2024.

[2] Rathbun et al. Sleepernets: Universal backdoor poisoning attacks against reinforcement learning agents. NeurIPS 2024.

[3] Ma et al. UNIDOOR: A Universal Framework for Action-Level Backdoor Attacks in Deep Reinforcement Learning. arXiv 2025.

**Ethical Concerns:**

["NO or VERY MINOR ethics concerns only"]

**Final Justification:**

I appreciate the authors' detailed responses. The authors have addressed most of my concerns and provided additional results, which I believe help improve the overall quality of the paper. However, due to my remaining concerns about the practical applicability of offline RL agents in end-to-end autonomous driving, as well as the limited novelty of the findings in this paper, I will maintain my original rating.

**Limitations:**

Could the authors derive more domain-specific and novel findings grounded in the AD scenarios? Some of the intuitions and observations in the paper have already been discussed in prior work. For example:

(1) The concept of trajectory-based triggers was introduced in [4].

(2) The key observation that “the target action can be falsely activated by similar but non-trigger behaviors” (Page 4, Line 170) was referred to as a “pseudo trigger” in [5]. Additionally, negative training strategy may make the backdoor sensitive to noise, where slight deviations can significantly affect backdoor activation. In real-world scenarios, various sources of interference cause sensor inaccuracies in vehicles, which means this strategy’s impact might not always be positive. Therefore, I recommend discussing this trade-off in more detail.

---
[4] Wang et al. BackdooRL: Backdoor attack against competitive reinforcement learning. IJCAI 2021.

[5] Guo et al. Policycleanse: Backdoor detection and mitigation for competitive reinforcement learning. ICCV 2023.

**Quality:**

2

**Strengths And Weaknesses:**

Strengths
- It is a meaningful attempt to explore backdoor attacks in autonomous driving.
- Using the behavioral trajectories of other agents as triggers is a meaningful and relevant approach.
- The proposed method is well-motivated and appears reasonable.
- The attack demonstration videos offer a helpful illustration of the proposed attack.

Weaknesses
- The writing requires further improvement for clarity.
- There is some mismatch between the threat model and the proposed method.
- The realism of the attack scenario warrants further discussion.

---

> ### Author Rebuttal · Authors · 2025-07-31
>
> We thank the reviewer for the questions and will incorporate all suggested changes in our next version.
>
> **Q1: Offline RL in AD.**
> Offline RL has been a core methodology in AD research, with studies demonstrating its performance [1], improving planning via expert knowledge [2], and deploying it in high-speed scenarios [3]. Dedicated benchmarks [4] have been introduced to evaluate offline RL driving agents. Safety-focused works [5, 6] study its reliability. Those works highlight that offline RL for AD is widely recognized and standardized.
>
> [1] Fang et al., Offline RL for AD with Real World Driving Data, ITSC'22.
>
> [2] Li et al., Boosting Offline RL for AD with Hierarchical Latent Skills, ICRA'24.
>
> [3] Pan et al., Agile AD using End-to-End Deep Imitation Learning, RSS'18.
>
> [4] Lee et al., AD4RL: AD Benchmarks for Offline RL with Value-based Dataset, ICRA'24.
>
> [5] Lin et al., Safety-Aware Causal Representation for Trustworthy Offline RL in AD, RA-L'24.
>
> [6] Shi et al., Offline RL for AD with Safety and Exploration Enhancement, ML4AD NeurIPS'21.
>
> **Q2: RL backdoor literature.**
> Due to space limits, we focus on data poisoning backdoors in DRL for AD, not general RL backdoor attacks. Below is a brief summary. Prior work explored backdoor attacks in single-agent DRL using state perturbation patches as triggers [1]. [2] introduces a two-agent setting, where the adversarial agent’s actions serve as the trigger. More recent work extends backdoors to multi-agent RL [3].
>
> [1] Kiourti et al., Trojdrl: Trojan attacks on DRL agents, DAC'20.
>
> [2] Wang et al., Backdoorl: Backdoor attack against competitive RL, IJCAI'21.
>
> [3] Chen et al., Marnet: Backdoor attacks against cooperative multi-agent RL, TDSC'22.
>
> **Q3: Use of third-party trajectory data, support of cited works for threat model.**
> Using third-party trajectory data to train driving agents is a common practice. [1] trains offline RL driving agents on Waymo’s trajectory data. [2] builds an imitation learner using human trajectories. [3] uses large-scale taxi trajectories to train a driving agent. We further clarify that our threat model does not directly assume training a driving agent on third-party trajectory data. The attacker is the data annotation service. Whether the AD developers use in-house or third-party data, they outsource the annotation step. Leveraging third-party annotation services is standard across AD. NuScenes[4] and PandaSet[5] use Scale AI for labeling trajectory data, BDD100K[6] uses Amazon Mechanical Turk. [7, 8] list annotation vendors used by AD companies. Thus, our threat model is fully aligned with practice. Regarding the cited works, while [22] and [60] are not directly AD-focused, [60] evaluates their attack on AD tasks, and [22] cites AD as a key motivation. [25] supports its threat model by noting AD developers’ use of third-party annotation services.
>
> [1] Rowe et al., CtRL-Sim: Reactive and Controllable Driving Agents with Offline RL, CoRL24.
>
> [2] Chen et al., Learning by cheating, CoRL'19.
>
> [3] https://wayve.ai/thinking/emerging-behaviour-of-our-driving-intelligence-with-end-to-end-deep-learning.
>
> [4] Caesar et al., nuScenes: A multimodal dataset for AD, CVPR'20.
>
> [5] Xiao et al., PandaSet: Advanced Sensor Suite Dataset for AD, ITSC'21.
>
> [6] BDD100K: A Diverse Driving Dataset for Heterogeneous Multitask Learning, CVPR'20.
>
> [7] Liu et al., A Survey on AD Datasets: Statistics, Annotation Quality, and a Future Outlook, IV'24.
>
> [8] M14 Intelligence. 2020. Autonomous Vehicle Data Annotation Market Analysis.
>
> **Q4: Attacker’s knowledge of the MDP.** In our setting, the attacker need not know the victim’s MDP in order to perform poisoning or target action manipulation. We directly modify different timesteps’ positions of the victim car within the training data, i.e., the attacker works entirely at the trajectory level and does not require the agent’s transition dynamics, reward, or action. For the trigger generation process, we leverage the TL framework to evaluate and keep those trajectories that satisfy our TL specifications. For target action generation, we directly perturb these coordinates to encode the target actions. Our pipeline operates at the coordinate level, so the adversary does not need to know the MDP.
>
> **Q5: Timing of poisoning.** We clarify that the poisoning does not run during the entire training process. We refine the configurable behavior models of surrounding attack vehicles until the generated trajectories satisfy our TL specifications. Then we deploy these finalized behavior models with the TL-guided parameter to collect the poisoning data. Finally, we inject the poisoning data into the full training set at a specific poisoning rate, without updating it.
>
> **Q6: Attacker as provider.** We agree with the reviewer that an attacker with access to both trajectory data and the MDP could typically be the agent provider. In this work, we follow existing studies and assume the attacker only needs access to the training data; thus we model them as third-party annotators. We agree that a provider‐level attack subsumes our scenario and will explicitly discuss in our revision that the adversaries can be the annotation vendor or the agent provider, depending on their available knowledge.
>
> **Q7: State/action clarification.** Our state space is a 259-dimensional vector containing sensor inputs and ego-car status. Action consists of low-level control commands: steering and throttle. We describe the state, action space in Appendix B, and will move key elements to Sec. 3 in our next version.
>
> **Q8: Lowering the poisoning rate (PR).** We select BadRL for its design focus on minimizing the PR. It uses a pre-trained Q-network to identify states where the target action is most harmful, and adds a mutual information regularizer. While our threat model assumes the attacker has no control over training, we relax this assumption to adapt BadRL and customize it for behavior cloning, which lacks a value function. Instead of using Q-value, we estimate the attack value using the log-probability gap between the optimal and target actions. We poison the top-k% highest-gap states, and modify the action to the target. We then introduce the mutual-information regularizer that maximizes the similarity between parameter-space gradients produced by triggered and clean versions of the same state. Tab. 1 shows that BadRL can help reduce 4% of the poisoning rate. We will include more analysis in the revision.
>
> **Tab. 1 Applying BadRL on BC (medium task).**
> PR(%)|Rew (ori / ben / poi)|ADE (ori / ben / poi)|MVR (ori / ben / poi)
> -|-|-|-
> 10(w/o BadRL)|180.30/183.59/34.77|0.72/2.14/70.21|0.42/0.45/0.71
> 8(w BadRL)|180.30/86.16/36.81|0.72/2.29/71.14|0.42/0.44/0.69
> 6(w BadRL)|180.30/185.43/45.38|0.72/2.18/75.48|0.42/0.41/0.66
>
> **Q9: Experimental structure.** In Sec. 4.2, we demonstrate our attack's effectiveness across 3 RL algorithms, various trigger patterns, and target actions, with detailed analysis on the robustness of RL algorithms and trigger patterns’ influence on attack effectiveness. We will follow the reviewer's suggestions to improve the flow.
>
> **Q10: OOD trajectory defense.** In Sec. 4.3, we evaluated our attack using a defense aimed at smoothing out OOD trajectories. We extend the defense with our ADE metrics: trajectories with ADE above a threshold (empirically set to 50) are removed. Tab. 2 shows that this defense mitigates the backdoor behavior, although it still underperforms a fully benign policy. This is likely because a single ADE threshold cannot filter out all poisoning trajectories, and more intelligent threshold selection is needed. We observe a drop in benign reward, possibly due to the removal of high-ADE trajectories that include complex traffic scenarios or failure cases, which improves the agent’s robustness. We believe this defense is promising, and we leave it as our future work.
>
> **Tab. 2 Performance comparison under ADE defense.**
> Task|Trigger|Poisoned Reward (w/o / w.)|Benign Reward (w/o / w.)
> -|-|-|-
> Easy|Sync-bypass|8.23/141.30|368.25/342.47
> ||Overtake|15.39/180.72|359.65/314.89
> Medium|Sync-bypass|42.58/150.76|309.67/288.54
> ||Overtake|38.39/166.03|299.83/261.31
>
> **Q11: Domain-specific insights & pseudo triggers (PT).** Our main contribution is not the trajectory trigger itself, but the introduction of complex, customizable multi-vehicle triggers. The proposed TL-driven procedure can automatically generate those trigger trajectories, which are not discussed in existing works. [5]'s PT differs from our setting as [5] presents a defense method where the PT is the detection agent’s trajectory that causes another agent to fail. While we act as the attacker, crafting stealthy backdoors that avoid unintended activations. For visual similarity, [5] notes that PT can diverge from the original trigger. Our “falsely trigger trajectories” are very similar to the true triggers. This precision guarantees that the target action is only activated under the specified trigger pattern.
>
> **Q12: Negative-training (NT) trade-offs.** To evaluate the trade-offs, we prepare three policies: a backdoored policy with and without NT, and a clean policy. We simulated realistic perception noise by adding zero-mean Gaussian noise to the ego car’s LiDAR points, whose default detection radius is 50m. We defined low, medium, and high noise levels with std of 5,10, and 15cm, and applied noise at every time step. In Tab. 3, the backdoor remains effective under all noise levels, denoted by the relatively lower poisoned reward. However, NT increases the policy’s sensitivity to noise, as the benign and poisoned rewards are all lower compared to the policy without NT.
>
> **Tab. 3 NT trade-offs.**
> Noise|Poisoned Reward (clean / w/o NT / w. NT)|Benign Reward (clean / w/o NT / w. NT)
> -|-|-
> No noise|379.18/10.47/7.12|385.09/371.09/368.25
> Low|358.22/8.96/9.96|355.41/352.27/345.54
> Medium|335.18/14.28/20.58|347.21/335.19/330.01
> High|304.76/60.17/80.08|319.16/305.83/297.63

---

> > ### Comment · Reviewer_AzS2 · 2025-08-06
> >
> > I appreciate the authors' detailed responses. The authors have addressed most of my concerns and provided additional results, which I believe help improve the overall quality of the paper. However, due to my remaining concerns about the practical applicability of offline RL agents in end-to-end autonomous driving, as well as the limited novelty of the findings in this paper, I will maintain my original rating.

---

> > > ### Author Response · Authors · 2025-08-06
> > >
> > > We appreciate the reviewer’s additional feedback and are pleased that our rebuttal addressed most of the concerns.
> > >
> > > **Practicality of offline RL in end-to-end autonomous driving.** As we have briefly discussed in our rebuttal, a large body of work, from application studies and benchmarks to safety-oriented extensions, demonstrates that offline RL has rapidly matured in end-to-end autonomous driving systems. We will expand our introduction to better motivate our attack and add those related works.
> > >
> > > **Novelty of our contributions**. To the best of our knowledge, we are the first to leverage temporal logic to specify diverse, physically realizable, multi-vehicle trajectory triggers for backdoor attacks in an end-to-end driving context. Moreover, we show our attack remains effective across multiple training algorithms, defenses, and highly diverse trigger patterns. We will strengthen our discussion of these points in the next revision.

---

### Official Review · Reviewer_eQXa · 2025-06-27

**Clarity:** 3
**Significance:** 2
**Originality:** 3
**Rating:** 5
**Confidence:** 2

**Summary:**

This paper focuses on exploring the vulnerability of end-to-end autonomous driving systems against backdoor attacks. While existing methods mainly rely on patch triggers, this work proposes a backdoor attack with triggers that are adaptable to deploy in the physical world, which is the trajectories of one or more attacker-controlled vehicles. A TL-based framework is used to generate sophisticated trajectories of different vehicles. Evaluations on different RL agents demonstrate the effectiveness and feasibility of the proposed framework.

**Questions:**

1. The proposed trajectory-based backdoor attacks assume there are other vehicles in the environment. What if there's not?
2. Can the poisoned trajectory be triggered by other components, like a pedestrian or bicyclist?

**Ethical Concerns:**

["NO or VERY MINOR ethics concerns only"]

**Final Justification:**

In my review, I mentioned my concerns about the diversity of trajectory-based triggers and some questions about whether the attack can be triggered by components other than vehicles, and the authors provided experiments to convince me. Hence, I think this paper is technically solid and should be accepted.

**Limitations:**

See weaknesses. No potential negative societal impact.

**Paper Formatting Concerns:**

No paper formatting concerns

**Quality:**

2

**Strengths And Weaknesses:**

Strengths:
1. I think the idea of trajectory-based backdoor attacks is interesting and practical. The motivation of this work is solid.
2. The paper is well-organized and easy to follow.
3. Video demos are provided on a webpage.

Weaknesses:
1. I don't think the trajectory-based backdoor attacks can generate a lot of different triggers. In the paper, three trigger trajectory patterns are shown. I think that is a disadvantage of trajectory-based backdoor attacks.
2. Minor weakness: The format of section heads should be consistent.

---

> ### Author Rebuttal · Authors · 2025-07-31
>
> We thank the reviewer for the questions and will incorporate all suggested changes in our next version.
>
> **Q1: The reviewer raised the concern about the diversity of trajectory-based triggers for the backdoor attack.**
>
> We thank the reviewer for raising this concern. Our temporal logic specification framework is inherently flexible and supports the definition of diverse trigger patterns. To address the reviewer’s concern, we design three additional trigger patterns: 1) one vehicle bypasses from one side while another suddenly brakes, 2) four vehicles bypass the ego car—two doing so synchronously, followed by another two synchronously, and 3) an asynchronous bypass, where two vehicles bypass the ego car with a 10-second interval. We evaluate these patterns on easy tasks using the Coptidice-trained agents, with a poisoning rate of 10%. As shown in in Tab. 1 below, our attack can support a diverse range of trigger designs. While all triggers are effective, the attack performance is slightly reduced for more complex patterns, such as the four-vehicle bypass. This is likely due to the increased state complexity introduced by multiple vehicles, which may require a higher poisoning rate to be fully effective. We will incorporate the results, analysis, and videos in our next version.
>
> **Tab. 1 Attack effectiveness of three new trigger patterns.**
>
> | Task | Trigger Pattern | Original Reward | Benign Reward| Poisoned Reward| Original ADE| Benign ADE| Poisoned ADE| Original MVR | Benign MVR| Poisoned MVR|
> |------|-----------------|-----------------|---------------|-----------------|--------------|------------|--------------|--------------|------------|--------------|
> | Easy | Bypass-brake    | 388.06          | 362.00        | 128.81          | 0.31         | 1.55       | 78.14        | 0.00         | 0.00       | 1.00         |
> | Easy | 4 bypass        | 388.06          | 350.18        | 148.96          | 0.31         | 1.63       | 65.35        | 0.00         | 0.03       | 0.86         |
> | Easy | Async-bypass    | 388.06          | 388.24        | 57.28           | 0.31         | 0.96       | 102.27       | 0.00         | 0.00       | 1.00         |
>
> Additionally, we present the results of single-vehicle trigger patterns in Tab. 11 in Appendix D.4, further demonstrating the expressiveness of our framework. That said, there is a trade-off between trigger complexity and stealthiness: overly complex triggers may be difficult for the model to learn and for the attacker to successfully inject, while overly simple triggers may occur frequently in everyday scenarios, making them more detectable.
>
> **Q2: The reviewer asked what would happen if no other vehicles are present in the environment, and whether a pedestrian or bicyclist could instead serve as the trigger.**
>
> We thank the reviewer for the question. We would like to clarify that our threat model assumes the attacker can deliberately and strategically control one or multiple vehicles and drive along specific, physically plausible trigger trajectories; these motions will be observed by the ego car’s LiDAR and activate the backdoor. The attacker’s vehicles respect safety rules and do not collide with the ego car. As a result, our trigger does not rely on ambient or “natural” traffic flow, and the attacker controls the interacting vehicles as needed. In the hypothetical case where no other vehicles are present and **the attacker also does not introduce one**, the backdoor cannot be activated, which is consistent with our model. Furthermore, to address concerns about accidental activation, we conduct experiments in Appendix D.1 to demonstrate that our designed trigger patterns are rare during a daily life driving scenario, indicating a low likelihood of false positives in natural traffic.
>
>
> Our poisoning does not depend on the type of triggers. To use bicyclists or pedestrians as triggers, we replace the default vehicle behavior model with one that captures their specific dynamics. MetaDrive already includes built-in dynamics models for bicyclists and pedestrians, so we simply swap in those models when generating trigger trajectories for these object types. For bicyclists, the simulator approximates the rider and bike as a single rigid‐body box whose motion is updated kinematically: at each simulation step, the engine assigns a desired linear velocity to the body, allowing it to follow any prescribed 2-D trajectory while still participating in collision checks of the simulator.  Pedestrians are treated identically, differing only in their collision envelope shape and dimensions. In our experiments, we use the “sync-bypass” as the trigger path, swapping out the vehicle as a bicyclist and a pedestrian, respectively, and training the agent with Coptidice. Results in Tab. 2 show that for bicyclists and pedestrians, the backdoor still triggers but causes slightly less significant reward loss compared to the vehicle. This is likely because bicyclists and pedestrians generate weaker occlusion and kinematic disruption than the vehicle. The benign driving is barely affected, denoted by the similar benign reward, ADE, and MVR, compared to the car-based trigger.
>
> **Tab. 2 Attack effectiveness of “sync-bypass” trigger when replacing vehicle with bicyclist and pedestrian.**
>
> | Task   | Trigger Type      | Original Reward| Benign Reward | Poisoned Reward| Original ADE| Benign ADE| Poisoned ADE| Original MVR| Benign MVR | Poisoned MVR |
> |--------|------------|-----------------|---------------|-----------------|--------------|------------|--------------|--------------|------------|--------------|
> | Easy   | Bicyclist  | 388.06          | 360.30        | 15.12           | 0.31         | 1.60       | 85.10        | 0.00         | 0.05       | 0.98         |
> | Easy   | Pedestrian | 388.06          | 358.04        | 25.23           | 0.31         | 1.65       | 70.76        | 0.00         | 0.08       | 0.85         |
> | Medium | Bicyclist  | 319.06          | 289.54        | 55.14           | 0.28         | 1.05       | 65.45        | 0.23         | 0.22       | 0.90         |
> | Medium | Pedestrian | 319.06          | 295.15        | 80.06           | 0.28         | 1.10       | 55.50        | 0.23         | 0.21       | 0.83         |
>
> **Q3: The reviewer pointed out the consistency in the format of section heads.**
>
> We thank the reviewer for pointing this out and will modify it in our next version.

---

> > ### Comment · Reviewer_eQXa · 2025-08-03
> >
> > Thank you for your rebuttal. My concerns have been addressed. Please incorporate the new experiments in the revised version. I would raise my score to Accept.

---

> > > ### Author Response · Authors · 2025-08-04
> > >
> > > Thank you for your thoughtful comment! We sincerely appreciate your time and feedback throughout the review process, and we will incorporate all new experiments in our next version.

---

### Official Review · Reviewer_hxgb · 2025-07-01

**Clarity:** 3
**Significance:** 2
**Originality:** 2
**Rating:** 5
**Confidence:** 4

**Summary:**

This paper proposes a trajectory-based backdoor attack against end-to-end AD systems, where existing literature mainly focus on pixel-level trigger and attack. The authors propose to use temporal logic formula to describe attaching vehicles' behavior and demonstrate the possibility of generating dynamic triggers. The authors conduct experiments and ablation study to validate the system design.

**Questions:**

Are the authors considering to include more diverse behavior of attacker vehicles, expressed by temporal logics? For example, traffic rules can be incorporated as well.

**Ethical Concerns:**

["NO or VERY MINOR ethics concerns only"]

**Final Justification:**

The authors have addressed most of my concerns.

**Limitations:**

Please see weaknesses.

**Paper Formatting Concerns:**

None.

**Quality:**

2

**Strengths And Weaknesses:**

Strengths:

1. I think it's novel to introduce temporal logic specifications to formally describe the attacker's behavior;
2. The paper studies a trajectory-based backdoor attack instead of pixel/image-level  attack, where most previous work study;
3. The paper looks clear and well-written to me.

Weaknesses:

1. The empirical study is still limited to simulating environments. The paper can be further strengthened if extending to real-world evaluation, where practical conditions (like weather) can affect the attack as well.
2. The assumption that the attacker can access the whole training data of ego vehicle is a bit strong, which may be challenging to obtain in practice.

---

> ### Author Rebuttal · Authors · 2025-07-31
>
> We thank the reviewer for the questions and will incorporate all suggested changes in our next version.
>
> **Q1: The reviewer raised the concern that the experiments are mainly conducted in a simulator, with no real-world evaluation, also suggested evaluating the robustness against different weather conditions.**
>
> We thank the reviewer for the insightful comment. We agree with the reviewer that testing on real-world vehicles will demonstrate the practical effectiveness. We do not perform real-world evaluations mainly because executing backdoor attacks’ target action in the real world, such as sudden left turns, raises serious safety concerns and could result in dangerous situations. Additionally, to the best of our knowledge, no prior work on backdoor attacks in AD has been demonstrated on real vehicles. We follow existing work setups [1, 2, 3] and conduct our experiments in the well-established simulator with closed-loop evaluation. Furthermore, deploying end-to-end offline RL in a production vehicle requires extensive engineering and still remains an active research challenge; thus, it is beyond this paper’s scope. Lastly, our simulator adheres to realistic physical constraints and traffic rules. For example, when vehicles are off the road, the simulator automatically terminates the trajectory. Thus, we believe that our simulator-based results sufficiently establish the attack’s feasibility and impact, and we leave real-vehicle testing to future work.
>
> To assess robustness under varying weather, we simulate fog and rain in the simulator. For fog, we impair LiDAR accuracy by randomly dropping 5% of the LiDAR points from the ego car’s state vector and setting those points to 0. For rain, we reduce tire-road friction by lowering the wheel friction parameter in the ego car’s dynamics model from the default value of 0.9 to 0.5. We evaluate in the medium tasks using BC and Coptidice trained agents, the target action is “sudden left turn” and the trigger pattern is “sync-bypass”, poisoning rate as 10%, following the same setup as Tab. 2 in the paper. The results are presented in Tab. 1 below. First, we observe that the noise will degrade the clean model’s performance. For backdoored policies, the injected trigger still remains effective, as evidenced by a clear performance gap between the benign and poisoned metrics. This demonstrates that our attack can still remain effective under different weather conditions, although the benign metrics are influenced due to the presence of the noise. All results are averaged over 100 trajectories. We will include these results in our next version.
>
> **Tab. 1 Attack effectiveness under two weather conditions on the medium task.**
>
> | Weather | Algorithm  | Original Reward| Benign Reward| Poisoned Reward| Original ADE| Benign ADE| Poisoned ADE| Original MVR| Benign MVR| Poisoned MVR|
> |---------|------------|------------------|----------------|------------------|---------------|-------------|----------------|----------------|--------------|----------------|
> | Fog     | BC         | 153.26           | 156.05         | 31.03            | 1.08          | 3.21        | 73.72          | 0.55           | 0.60         | 0.74           |
> |         | Coptidice  | 271.20           | 253.22         | 40.45            | 0.42          | 2.90        | 83.32          | 0.30           | 0.39         | 0.89           |
> | Rain    | BC         | 117.19           | 119.33         | 22.60            | 1.80          | 4.79        | 75.73          | 0.68           | 0.73         | 0.78           |
> |         | Coptidice  | 207.39           | 201.28         | 29.68            | 0.70          | 2.33        | 85.01          | 0.40           | 0.45         | 1.00           |
> | Normal  | BC         | 180.30           | 183.59         | 34.77            | 0.72          | 2.14        | 70.21          | 0.42           | 0.45         | 0.71           |
> |         | Coptidice  | 319.06           | 309.67         | 42.58            | 0.28          | 0.93        | 83.32          | 0.23           | 0.15         | 1.00           |
>
> [1] Gong et al., Baffle: Hiding backdoors in offline reinforcement learning datasets, S&P'24.
>
> [2] Pourkeshavarz et al., Adversarial Backdoor Attack by Naturalistic Data Poisoning on Trajectory Prediction in Autonomous Driving, CVPR'24.
>
> [3] Han et al., Physical backdoor attacks to lane detection systems in autonomous driving, ACM ICM'22.
>
> **Q2: The reviewer raised the concern about the practicability of our threat model, having access to the whole training data can be challenging.**
>
> We thank the reviewer for the question. Our threat model assumes that the attacker is the third-party data annotation service, an entity to which the AD developers outsource the labeling process of the trajectory data. This setup grants the attacker full and reasonable access to the training data.
>
> Leveraging third-party annotation services (e.g., Scale AI, Amazon Mechanical Turk, etc.) is a well-established and widely adopted practice in the AD field. For example, leading datasets such as nuScenes [1] and PandaSet [2] explicitly credit Scale AI for labeling trajectory data, while BDD100K [3] leverages Amazon Mechanical Turk. Both academic surveys [4] and empirical industry studies [5] document an entire ecosystem of external annotation vendors that AD companies rely on. Thus our assumption that the attacker is from the third-party annotation service with access to the whole training data is consistent with real-world practice. Furthermore, we assume the attacker has access only to the data, not the training algorithm, which is fully aligned with our defined threat model about the data annotation service.
>
> [1] Caesar et al., nuScenes: A multimodal dataset for autonomous driving, CVPR'20.
>
> [2] Xiao et al., PandaSet: Advanced Sensor Suite Dataset for Autonomous Driving, ITSC'21.
>
> [3] BDD100K: A Diverse Driving Dataset for Heterogeneous Multitask Learning, CVPR'20.
>
> [4] Liu et al., A Survey on Autonomous Driving Datasets: Statistics, Annotation Quality, and a Future Outlook, Intelligent Vehicles 2024.
>
> [5] M14 Intelligence. 2020. Autonomous Vehicle Data Annotation Market Analysis, https://m14intelligence.com/public/index.php/product/11.
>
>
> **Q3: The reviewer suggested considering more diverse behavior of attacker vehicles, e.g., incorporating traffic rules into the temporal logic specifications.**
>
> We thank the reviewer for the insightful suggestion. Following the reviewer’s suggestion, we consider two traffic rules, one is speed limits, another is lane keeping. Specifically, alongside the existing atomic proposition that captures vehicle position, we introduce two new atomic propositions for speed and lane membership:
> (1) $\mu_{\text{speed}}^{i}(t,v_{\max}) \:\Leftrightarrow\ v_i(t)\le v_{\max}$ (2) $\mu_{\text{lane}}^{i}(t,\mathcal{L}) \:\Leftrightarrow\ (x_i(t),y_i(t))\in\mathcal{L}$.
> where $\mathcal{L}$ is the polygonal region of the current lane.
> We add them to our existing temporal logic predicates and define two new traffic-rule constraints directly within our propositions.
>
> $\texttt{SpeedLimit}(v_{\max},[t_s,t_e]) := \mathbf{G}{[t_s,t_e]}\,\mu_{\text{speed}}^{i}(t,v_{\max})$
>
> $\texttt{LaneKeep}(\mathcal{L},[t_s,t_e]) := \mathbf{G}{[t_s,t_e]}\,\mu_{\text{lane}}^{i}(t,\mathcal{L})$
>
> We use Coptidice and extend the traffic rules on our sync-bypass behavior, meaning that when the attacker performs this, it strictly follows the speed limits or lane keeping. The target action is to brake suddenly. We set the speed limit as 60mph. Results from Tab. 2 below show that incorporating traffic rules into the attacker's trajectory still yields effective attack performance, demonstrating that our framework is flexible and supports the definition of common traffic rules into attacker behavior. All results are averaged over 100 trajectories.
>
> **Tab. 2 Attack effectiveness of two new trigger patterns incorporating traffic rule compliance.**
>
> | Task   | Traffic Rule |Original Reward| Benign Reward| Poisoned Reward| Original ADE| Benign ADE| Poisoned ADE | Original MVR| Benign MVR| Poisoned MVR|
> |--------|--------------|-----------------|---------------|-----------------|--------------|------------|--------------|--------------|------------|--------------|
> | Easy   | SpeedLimit        | 388.06          | 372.31        | 12.97           | 0.31         | 1.51       | 100.68       | 0.00         | 0.00       | 1.00         |
> | Easy   | LaneKeep         | 388.06          | 365.20        | 10.06           | 0.31         | 1.48       | 103.75       | 0.00         | 0.00       | 0.98         |
> | Medium | SpeedLimit        | 319.06          | 310.17        | 39.28           | 0.28         | 0.92       | 94.54        | 0.23         | 0.11       | 1.00         |
> | Medium | LaneKeep         | 319.06          | 305.05        | 44.62           | 0.28         | 1.21       | 85.10        | 0.23         | 0.17       | 0.99         |

---

> > ### Comment · Reviewer_hxgb · 2025-08-04
> > **Response's to the rebuttal**
> >
> > Thank you for your rebuttal and I am less concerned now. I will raise my score to accept.

---

> > > ### Author Response · Authors · 2025-08-04
> > >
> > > Thank you for your kind comment! We’re truly grateful that our rebuttal addressed your concerns, and we appreciate the time and thoughtful feedback you provided throughout the review process.

---

### Decision · Program_Chairs · 2025-09-17

**Decision:**

Accept (poster)

**Comment:**

This work proposes a backdoor attack on autonomous driving to influence the vehicle's trajectory, assuming that data is labeled by a third-party (untrusted) service.
The main weaknesses remaining after the rebuttal are the following (quoting one of the reviewers):
- End-to-End Assumption: It's a simplification, as real-world AV systems are more complex and modular.
- Poisoning Rate: The 5%+ poisoning rate is unrealistic in practice but is acceptable for demonstrating the attack concept.

Nevertheless, these are minor points, and all reviewers are unanimous in recommending acceptance for this paper.